# VolcanoSV enables accurate and robust structural variant calling in diploid genomes from single-molecule long read sequencing

Can Luo[1,4], Yichen Henry Liu[2,4] & Xin Maizie Zhou [1,2,3] ✉

Structural variants (SVs) significantly contribute to human genome diversity and play a crucial role in precision medicine. Although advancements in single-molecule long-read sequencing offer a groundbreaking resource for SV detection, identifying SV breakpoints and sequences accurately and robustly remains challenging. We introduce VolcanoSV, an innovative hybrid SV detection pipeline that utilizes both a reference genome and local de novo assembly to generate a phased diploid assembly. VolcanoSV uses phased SNPs and unique $k$-mer similarity analysis, enabling precise haplotype-resolved SV discovery. VolcanoSV is adept at constructing comprehensive genetic maps encompassing SNPs, small indels, and all types of SVs, making it well-suited for human genomics studies. Our extensive experiments demonstrate that VolcanoSV surpasses state-of-the-art assembly-based tools in the detection of insertion and deletion SVs, exhibiting superior recall, precision, F1 scores, and genotype accuracy across a diverse range of datasets, including low-coverage (10x) datasets. VolcanoSV outperforms assembly-based tools in the identification of complex SVs, including translocations, duplications, and inversions, in both simulated and real cancer data. Moreover, VolcanoSV is robust to various evaluation parameters and accurately identifies breakpoints and SV sequences.

Structural variants (SVs) refer to alterations in the genomic DNA that are greater than 50 base pairs (bp), encompassing insertions, deletions, translocations, duplications, and inversions[1]. Even though the absolute number of SVs in a typical human genome is less than small variants like single nucleotide polymorphism (SNPs), due to their large size, SVs impact more base pairs than all single-nucleotide differences in the human genome[2]. Consequently, SVs have a significant impact on individual phenotypes and diseases[3–5], such as Parkinson's diseases[6]. Moreover, a lot of cancers are proven to be related to somatic SVs[7–12]. Therefore, the accurate and comprehensive characterization of SVs is of great clinical significance. However, SV characterization still remains one of the least resolved problems in genomics[13].

Before the advent of long-read sequencing technologies, SV detection methods primarily relied on short-read sequencing (next-generation sequencing). These methods often depended on discordant read pairs, read depth analysis, split-read alignments, and de novo assembly[14]. However, due to the limited length of short reads and the absence of extensive genomic context, SV detection using short reads typically struggled to achieve high recall or precision, particularly for large insertions.

The introduction of single-molecule sequencing, exemplified by PacBio and Nanopore, has been an important development for improving SV detection[15]. Long reads, with average lengths ranging from 10 to 20 kilobases (kb), offer the capability to readily span

[1]Department of Biomedical Engineering, Vanderbilt University, Nashville, TN, USA. [2]Department of Computer Science, Vanderbilt University, Nashville, TN, USA. [3]Data Science Institute, Vanderbilt University, Nashville, TN, USA. [4]These authors contributed equally: Can Luo, Yichen Henry Liu.
✉e-mail: maizie.zhou@vanderbilt.edu

medium to large SVs without the need for complex assembly processes. This enables the detection of SVs by analyzing mapping information alone. The drawback of traditional long reads is their error-prone sequencing, i.e., the error rate of traditional long reads ranges from 10% to 20%. Hence, error correction and false discovery control are necessary for tools using traditional long reads. A new technology, PacBio Hifi reads, can achieve high accuracy (comparable to short reads accuracy, <1% error rate) while keeping the advantage of expanding large genomic regions. Hifi reads achieve high accuracy by using multiple subreads and sequencing by consensus result. Using Hifi reads, SV detection tools can avoid the need for error correction, enhance efficiency, and gain a more reliable result at the same time.

The standard approach for analyzing whole-genome long-read data from an individual involves aligning it to a reference genome (read alignment-based) to detect variants. Read alignment-based methods are attractive because they do not require extensive computational resources or high sequencing coverage. Numerous read alignment-based SV callers have emerged recently, including NanoVar[16], cuteSV[17], pbsv[18], Sniffles2[19], MAMnet[20], SVDSS[21], and DeBreak[22]. Classifying a tool as alignment-based is not absolute, even though it utilizes read alignment. For instance, DeBreak can be considered a hybrid method as it combines read alignment with local assembly to detect SVs. Similarly, MAMnet can be categorized as a deep learning-based approach that relies on read alignment information. Hybrid and deep learning-based methods are promising in detecting SVs compared to traditional alignment-based methods in most aspects. Nevertheless, methods relying even partially on reads alignment often have limitations in the accurate representation of a complete genome, an SV's initial and ending position in the genome (its "breakpoints"), and in identifying the full SV sequence[22]. Alignment-based, and these hybrid or deep learning-based methods concentrate solely on target and local signals, making it challenging to generate a comprehensive map for genome-wide variants.

An alternative strategy is to assemble the entire genome of an individual solely based on their reads (de novo assembly) and then compare the assembly with a reference genome, a process that demands more computational resources. Currently, only a handful of assembly-based SV callers have been introduced, such as Dipcall[23], SVIM-asm[24], and PAV[25]. Assembly-based methods surpass alignment-based methods in achieving accurate SV detection with respect to breakpoints and SV sequences. Nevertheless, whole-genome assembly often produces a complete map for genome-wide variants by compromising precision and incurring a significant number of false positives, particularly in the case of error-prone long reads.

Here, to address existing limitations in alignment-based and assembly-based methods, we present VolcanoSV, an innovative hybrid SV detection pipeline that leverages phased SNPs to generate phased diploid assembly for precise haplotype-resolved SV analysis. VolcanoSV offers several advantageous features. (1) It outperforms state-of-the-art assembly-based SV callers, demonstrating higher recall, precision, F1 scores, and genotype accuracy across a diverse range of datasets, including low-coverage (10x) datasets, without compromising accuracy. (2) VolcanoSV is compatible with all mainstream long-read sequencing platforms, which vary considerably in sequencing error rates, and can discover various types of SVs including deletions, insertions, duplications, inversions, and translocations. Moreover, VolcanoSV excels in detecting and phasing SNPs and small indels along with SVs. (3) It exhibits more robust performance by accurately identifying breakpoints and SV sequences. (4) VolcanoSV-vc, the assembly-based SV calling component, has a low false discovery rate. With these features, VolcanoSV is well-suited for generating comprehensive genetic maps for human genomics studies.

## Results

We first investigated SV detection using 4 assembly-based methods (VolcanoSV (v1.0.0), PAV (freeze2), SVIM-asm (v1.0.2), and Dipcall) in 14 PacBio Hifi, CLR, and ONT datasets, 9 simulated long reads datasets, and two paired tumor-normal CLR and ONT datasets. For Hifi data, three assembly-based SV callers (PAV, SVIM-asm, and Dipcall) could use as input the diploid assembly result of hifiasm (v0.16)[26]. For CLR and ONT data, Flye (v2.9-b1768)[27] plus HapDup (v0.5-iss10)[28] were used to generate a dual assembly for the three assembly-based tools. We selected hifiasm and Flye plus HapDup to generate assembly since they provided the best assembly results for SV calling[29]. VolcanoSV employed its own haplotype-aware assembly component (VolcanoSV-asm) to produce a diploid assembly. To further demonstrate VolcanoSV's robust performance across different SV evaluation thresholds, we compared SV calls from four assembly-based methods in terms of breakpoint identification and SV sequence accuracy. Among the 14 long-read sequencing datasets (Table 1), five PacBio HiFi datasets were referred to as Hifi_L1, Hifi_L2, Hifi_L3, Hifi_L4, and Hifi_L5. They had approximately 56×, 30×, 34×, 28×, and 41× coverage, respectively. Three PacBio CLR datasets were referred to as CLR_L1, CLR_L2, and CLR_L3 and their coverage was 89x, 65x, and 29x, respectively. We also used six ONT datasets referred to as ONT_L1, ONT_L2, ONT_L3, ONT_L4, ONT_L5, and ONT_L6. Their coverage was approximately 48×, 46×, 57×, 36×, 47×, and 51×. More information for each SV caller and dataset is provided in Table 1. VolcanoSV utilizes a reference genome and long-read data to generate a high-quality haplotype-resolved diploid assembly. Using this assembly, all types of variants are comprehensively detected. The VolcanoSV pipeline is illustrated in Figs. 1 and 2. A detailed description is provided in the Methods section.

### VolcanoSV exhibits exceptional performance across 14 real long-read datasets

To assess the performance of insertion and deletion SV detection, we applied four assembly-based tools, VolcanoSV, PAV, SVIM-asm, and Dipcall, across 14 long-read libraries of HG002. We evaluated their results against the Genome in a Bottle (GIAB) SV gold standard[23]. An SV benchmarking tool, Truvari (v4.0.0)[30], was employed to compare the SV calls of each tool with the GIAB SV gold standard. Truvari evaluates SVs within Variant Call Format files (VCF) by analyzing four essential similarity metrics (reference distance, reciprocal overlap, size similarity, sequence similarity) across all SV pairs within a designated region, while also ensuring SV type and genotype match between compared SV pairs. If any of these metrics exceed user-defined thresholds, the SV pair fails to be a candidate match. The following metric/parameter setting was used in Truvari: $p = 0.5, P = 0.5, r = 500, O = 0.01$, representing a moderate-tolerance metric/parameter set to evaluate SV calls. Specifically, parameter $p$, or pctstim, ranging from 0 to 1.0, controls the minimum allele sequence similarity used to identify the SV calls being compared as the same. Parameter $P$, also known as pctsize, ranges from 0 to 1.0, defining the minimum allele size similarity required between the two SVs. Parameter $O$, also referred to as pctovl, ranges from 0 to 1.0 and determines the minimum threshold of reciprocal overlap ratio between the base and comparison call. It is only applicable on deletions that can be used to evaluate their breakpoint shift. Parameter $r$, or refdist, ranging from 0 to 1000 bp, limits the threshold for maximum reference location difference of the SVs being compared, which can be used to evaluate the breakpoint shift of insertions.

We first determined the average performance across different PacBio Hifi, CLR, and ONT datasets for the four assembly-based tools (Table 2). Across Hifi datasets, VolcanoSV achieved the best average F1 (91.03% and 94.19%) and genotyping accuracy (98.32% and 99.01%) for insertions and deletions. Across CLR datasets, VolcanoSV achieved the best average F1 (89.72% and 93.70%) and genotyping accuracy (97.07%

**Table 1 | Resource for different tools and long-read datasets**

| Tools | Version | Resource | Variant Types | Link |
|---|---|---|---|---|
| Dipcall | 0.3 | Li et al.[23] | DEL, INS, small indel, SNP | https://github.com/lh3/dipcall |
| SVIM-asm | 1.0.2 | Heller et al.[24] | DEL, INS, DUP, INV, TRA | https://github.com/eldariont/svim-asm |
| PAV | 2.0.0 | Ebert et al.[25] | DEL, INS, INV, small indel, SNP | https://github.com/EichlerLab/pav |
| VolcanoSV | 1.0.0 | This paper[56] | DEL, INS, INV, TRA, DUP, small indel, SNP | https://github.com/maiziezhoulab/VolcanoSV |
| SVIM | 1.4.2 | Heller et al.[49] | DEL, INS, DUP, INV, TRA | https://github.com/eldariont/svim |
| cuteSV | 1.0.11 | Jiang et al.[17] | DEL, INS, DUP, INV | https://github.com/tjiangHIT/cuteSV |
| NanoVar | 1.3.9 | Tham[16] | DEL, INS, DUP, INV, TRA | https://github.com/benoukraflab/NanoVar |
| pbsv | 2.6.2 | | DEL, INS, DUP, INV, TRA, CNV | https://github.com/PacificBiosciences/pbsv |
| DeBreak | 1.0.2 | Yu et al.[22] | DEL, INS, DUP, INV, TRA | https://github.com/Maggi-Chen/DeBreak |
| Sniffles2 | 2.0.6 | Smolka et al.[19] | DEL, INS, DUP, INV, TRA | https://github.com/fritzsedlazeck/Sniffles |

| Dataset | Abbreviation in the paper | Coverage | Source |
|---|---|---|---|
| HG002 PacBio CCS 15kb+20kb | Hifi_L1 | 56× | https://ftp-trace.ncbi.nlm.nih.gov/ReferenceSamples/giab/data/AshkenazimTrio/HG002_NA24385_son/PacBio_CCS_15kb_20kb_chemistry2/reads/ |
| HG002 PacBio CCS 10kb | Hifi_L2 | 30× | https://ftp-trace.ncbi.nlm.nih.gov/ReferenceSamples/giab/data/AshkenazimTrio/HG002_NA24385_son/PacBio_CCS_10kb/ |
| HG002 PacBio CCS 11kb | Hifi_L3 | 34× | https://www.ncbi.nlm.nih.gov/sra/SRR8833180 |
| HG002 PacBio CCS 15kb | Hifi_L4 | 28× | https://ftp-trace.ncbi.nlm.nih.gov/ReferenceSamples/giab/data/AshkenazimTrio/HG002_NA24385_son/PacBio_CCS_15kb/ |
| HG002 PacBio CCS 16kb | Hifi_L5 | 41× | https://www.ncbi.nlm.nih.gov/bioproject/PRJNA832505 |
| CHM13 PacBio CCS SRR1129212 | Hifi_L6 | 33× | https://www.ncbi.nlm.nih.gov/bioproject/PRJNA530776 |
| CHM13 PacBio CCS SRX5633451 | Hifi_L7 | 23.7× | https://www.ncbi.nlm.nih.gov/sra/SRX5633451 |
| HG002 PacBio CLR | CLR_L1 | 89× | https://www.ncbi.nlm.nih.gov/sra/SRX7668835 |
| HG002 MtSinai | CLR_L2 | 65× | https://ftp-trace.ncbi.nlm.nih.gov/ReferenceSamples/giab/data/AshkenazimTrio/HG002_NA24385_son/PacBio_MtSinai_NIST/ |
| HG002 PacBio CLR | CLR_L3 | 29× | https://www.ncbi.nlm.nih.gov/sra/SRX6719924 |
| HG002 Nanopore PRJNA678534 | ONT_L1 | 48× | https://www.ncbi.nlm.nih.gov/Traces/study/?acc=SRP292617&o=acc_s%3Aa |
| HG002 Nanopore Promethion | ONT_L2 | 46× | https://ftp-trace.ncbi.nlm.nih.gov/ReferenceSamples/giab/data/AshkenazimTrio/HG002_NA24385_son/UCSC_Ultralong_OxfordNanopore_Promethion/ |
| HG002 Nanopore UL guppy3.2.4 | ONT_L3 | 57× | https://ftp.ncbi.nlm.nih.gov/ReferenceSamples/giab/data/AshkenazimTrio/HG002_NA24385_son/Ultralong_OxfordNanopore/guppy-V3.2.4_2020-01-22/ |
| HG002 Nanopore PAD64459 | ONT_L4 | 36× | https://s3-us-west-2.amazonaws.com/human-pangenomics/NHGRI_UCSC_panel/HG002/hpp_HG002_NA24385_son_v1/nanopore/HG002_ONT_PAD64459_Guppy_3.2.fastq.gz |
| HG002 Nanopore Standard_Unsheared_UCSC | ONT_L5 | 47× | https://s3-us-west-2.amazonaws.com/human-pangenomics/NHGRI_UCSC_panel/HG002/hpp_HG002_NA24385_son_v1/nanopore/downsampled/standard_unsheared/HG002_ucsc_Jan_2019_Guppy_3.4.4.fastq.gz |
| HG002 Nanopore UCSC_lt100kb | ONT_L6 | 51× | https://s3-us-west-2.amazonaws.com/human-pangenomics/NHGRI_UCSC_panel/HG002/hpp_HG002_NA24385_son_v1/nanopore/downsampled/greater_than_100kb/HG002_ucsc_ONT_lt100kb.fastq.gz |

**Table 1 (continued) | Resource for different tools and long-read datasets**

| Tools | Version | Resource | Variant Types | Link |
|---|---|---|---|---|
| 9 Simulated long reads datasets | | Hifi_TRA Hifi_DUP Hifi_INV CLR_TRA CLR_DUP CLR_INV ONT_TRA ONT_DUP ONT_INV | 40× | CLR&ONT: VISOR+PBSIM3; Hifi: VISOR+PBSIM3+CCS |
| HCC1395 Tumor PacBio | | HCC1395_PB | 39× | https://www.ncbi.nlm.nih.gov/sra/?term=SRR8955953 |
| HCC1395 Normal PacBio | | HCC1395BL_PB | 44× | https://www.ncbi.nlm.nih.gov/sra/?term=SRR8955954 |
| HCC1395 Tumor ONT | | HCC1395_ONT | 12× | https://www.ncbi.nlm.nih.gov/sra/?term=SRR16005301 |
| HCC1395 Normal ONT | | HCC1395BL_ONT | 19× | https://www.ncbi.nlm.nih.gov/sra/?term=SRR17096031 |

Top panel: The SV calling tools used in this paper. Tool version number, cited article, variant types called by each tool, and tool links are shown in the table. Bottom panel: The long-read datasets used in this paper. The abbreviation, coverage, and source link for each dataset are shown in the table. The abbreviation is used to refer to each dataset in the main text and supplementary information. For simulated datasets, the corresponding simulators are listed in the source field.

and 98.58%) for insertions and deletions. In ONT datasets, VolcanoSV also attained the best average F1 (90.10% and 93.13%), and genotyping accuracy (98.00% and 99.06%) for insertions and deletions.

When we examined each dataset (Fig. 3, Table 2, and Supplementary Tables 1–3), VolcanoSV consistently outperformed all other tools, achieving the highest F1 scores for both insertions and deletions for all 14 libraries. In the context of the five Hifi datasets (Fig. 3, Table 2, and Supplementary Table 1), VolcanoSV achieved the highest ranking in terms of all performance metrics. Specifically, for insertions, VolcanoSV consistently surpassed all other tools across all metrics, with F1 score, recall, precision, and GT concordance outperforming the second-ranked tool by an average of 1.29%, 0.67%, 1.92%, and 0.59%, respectively. With respect to deletions, VolcanoSV sustained its advantage, demonstrating an average superiority of 1.07% in F1 score, 0.48% in recall, 1.52% in precision, and 0.53% in GT concordance over the second-ranked tool.

In the three CLR datasets (Fig. 3, Table 2, and Supplementary Table 2), VolcanoSV stood out as the top performer across all metrics and libraries, with distinct advantages. For insertions, VolcanoSV's performance metrics, including F1 score, recall, precision, and GT concordance, were 3.30%, 0.87%, 4.61%, and 4.20% higher than the second-ranked tool. Likewise, for deletions, VolcanoSV outperformed the second-ranked tool by 4.87%, 6.19%, 3.19%, and 1.71% higher scores on average for F1, recall, precision, and GT concordance. It is noteworthy that CLR data exhibited a significantly higher error rate, varying between 10% to 20%. PAV, SVIM-asm, and Dipcall exhibited significantly inferior performance in PacBio CLR when contrasted with Hifi datasets. Effectively eliminating false positive calls is a crucial step following the SV detection process. VolcanoSV incorporates a precise SV filtering procedure and an advanced GT prediction model within its workflow, resulting in a notable enhancement in performance compared to all other tools.

For the six ONT datasets (Fig. 3, Table 2, and Supplementary Table 3), VolcanoSV still maintained a substantial lead. In terms of insertions, VolcanoSV outperformed the second-ranked tool, with an average F1 score and precision that were 1.5% and 2.68% higher, respectively. Regarding the insertion recall, on ONT_L3-5, VolcanoSV's recall was 0.38% higher on average than the second-ranked tool. On ONT_L1 and L6, VolcanoSV exhibited the second-highest recall, with just an average of 0.14% less compared to the top recall. However, on ONT_L2, VolcanoSV only demonstrated the third-highest recall, with 1.03% less compared to the top recall. For the insertion GT concordance, VolcanoSV achieved a 0.29% higher GT concordance than the second-ranked tool, except for ONT_L2, where SVIM-asm reached the highest GT concordance with a margin of 0.16% over VolcanoSV. In terms of deletions, VolcanoSV outperformed the competition with an average F1 score, precision, and GT concordance that were 0.89%, 1.87%, and 0.62% higher, respectively, than the second-ranked tool. In terms of deletion recall, on ONT_L1, L3, L4, and L6, VolcanoSV's recall was 0.13% higher on average than the second-ranked tool. However, on ONT_L2 and L5, PAV and SVIM-asm attained the best recall, with an increase of 0.18% on average, in comparison to VolcanoSV.

In summary, VolcanoSV emerged as the top-tier choice for assembly-based SV detection across different long-read datasets, exhibiting superior performance and consistency, particularly for PacBio HiFi and CLR datasets in terms of F1 score, recall, precision, and GT concordance. VolcanoSV still demonstrated its superiority for ONT datasets in terms of F1 score, precision, and GT concordance. With respect to the recall in insertions and deletions, VolcanoSV achieved the best recall in 3–4 out of 6 datasets.

## SV annotation

Although VolcanoSV further improved F1 scores and genotyping accuracy compared to the second-ranked tool in each dataset, the potential benefits impact of this improvement on downstream

## VolcanoSV Overall Pipeline

**Fig. 1 | VolcanoSV overall workflow.** The main workflow of VolcanoSV consists of two key components, VolcanoSV-asm and VolcanoSV-vc. VolcanoSV-asm (left, blue square) comprises three conceptual modules to perform diploid assembly (partitioning reads into corresponding haplotypes, assigning unphased reads, and performing a haplotype-aware local assembly). The output of this component is processed by the VolcanoSV-vc component (center, red rectangle) to perform variant detection. Further details are provided in the Methods section.

analyses remained unclear. To delve deeper into these unique true positive (TP) SVs with correct genotypes (GT) identified by VolcanoSV compared to the second-ranked tool, we annotated the SVs with their predicted effects on other genomic features using the Ensembl Variant Effect Predictor (VEP)[31].

For example, we extracted and annotated 300 unique TP SVs with correct genotypes identified by VolcanoSV compared to the second-ranked tool, SVIM-asm, in Hifi_L1. These SVs overlapped with 258 genes, 939 transcripts, and 120 regulatory features, including 266 novel SVs and 34 existing ones. Among these, 125 SVs were coding sequence variants and 16 were in-frame deletions. More information on the consequences of these SVs is demonstrated in Supplementary Fig. 1. Additionally, we performed an exact match comparison of these SVs in terms of sequence and breakpoints with those in the Genome Aggregation Database (gnomAD)[32]. We identified 29 matching SVs, 12 of which are rare variants with an allele frequency (AF) of less than 1%. Furthermore, we found that these SVs allowed an additional 85 genes to be phased by VolcanoSV. Genes are considered phased if all heterozygous variants within it are phased.

We performed these analyses for all 14 long-read datasets (Supplementary Table 4 and Supplementary Fig. 1), and the results suggested that the improvement by VolcanoSV over the second-ranked tool in each dataset provided a substantial number of SVs with potential phenotypic effects.

### VolcanoSV unveils numerous unique true structural variants and remains robust across SV sizes

As we observed in the previous section, VolcanoSV achieved the highest recall in all PacBio Hifi and CLR datasets, and half of the ONT datasets. To further assess the overlapping calls among different tools and understand why VolcanoSV generated many more true positive calls, we employed an UpSet plot to visualize the total number of shared true positive (TP) calls and unique TP calls for all four assembly-based tools (Fig. 4a–c). Overall, the four tools we examined exhibited a substantial overlap in TP calls, as indicated by the rightmost bar in the plot. Specifically, across Hifi_L2, CLR_L1, and ONT_L1 data, these four tools shared 8457, 7834, and 8151 TP calls, respectively. Notably, VolcanoSV contributed the largest number of unique TP calls. For Hifi_L2, VolcanoSV, PAV, SVIM-asm, and Dipcall had 43, 35, 4, and 1 unique TP calls, respectively. In the case of CLR_L1, these tools had 174, 28, 11, and 4 unique TP calls, while for ONT_L1, they had 56, 19, 5, and 2 unique TP calls. The SV annotation analysis for the unique SVs by VolcanoSV was illustrated in Supplementary Table 5 and Supplementary Fig. 2.

To assess the influence of structural variant (SV) size on the accuracy of SV detection, we generated F1 scores for SV detection across different SV size ranges (Fig. 4d–f). In general, VolcanoSV demonstrated top-tier accuracy across most size ranges, except for insertions within the 10–50 kb range. Specifically, in Hifi_L2, VolcanoSV consistently excelled in medium and large-size deletions (50 bp–6 kb) and insertions (50 bp–4 kb). When it came to very large insertions (6–50 kb), VolcanoSV's performance exhibited some variability. However, in the case of very large deletions (6–50 kb), VolcanoSV consistently maintained top performance, with the exception of deletions within the 6–7 kb range. Notably, in the case of extremely large deletions (10–50 kb), VolcanoSV outperformed other tools significantly in terms of F1 scores.

The performance of VolcanoSV was even more remarkable for CLR and ONT data. On CLR_L1 and ONT_L1, VolcanoSV achieved the highest F1 scores for deletions across all size ranges and insertions within the 50 bp–5 kb range. However, for very large insertions (5–50 kb), VolcanoSV's performance showed some variability. It is worth highlighting that in CLR data, VolcanoSV displayed notably superior performance in detecting very large deletions (7–50 kb) compared to other tools. Upon further analysis of the remaining 11

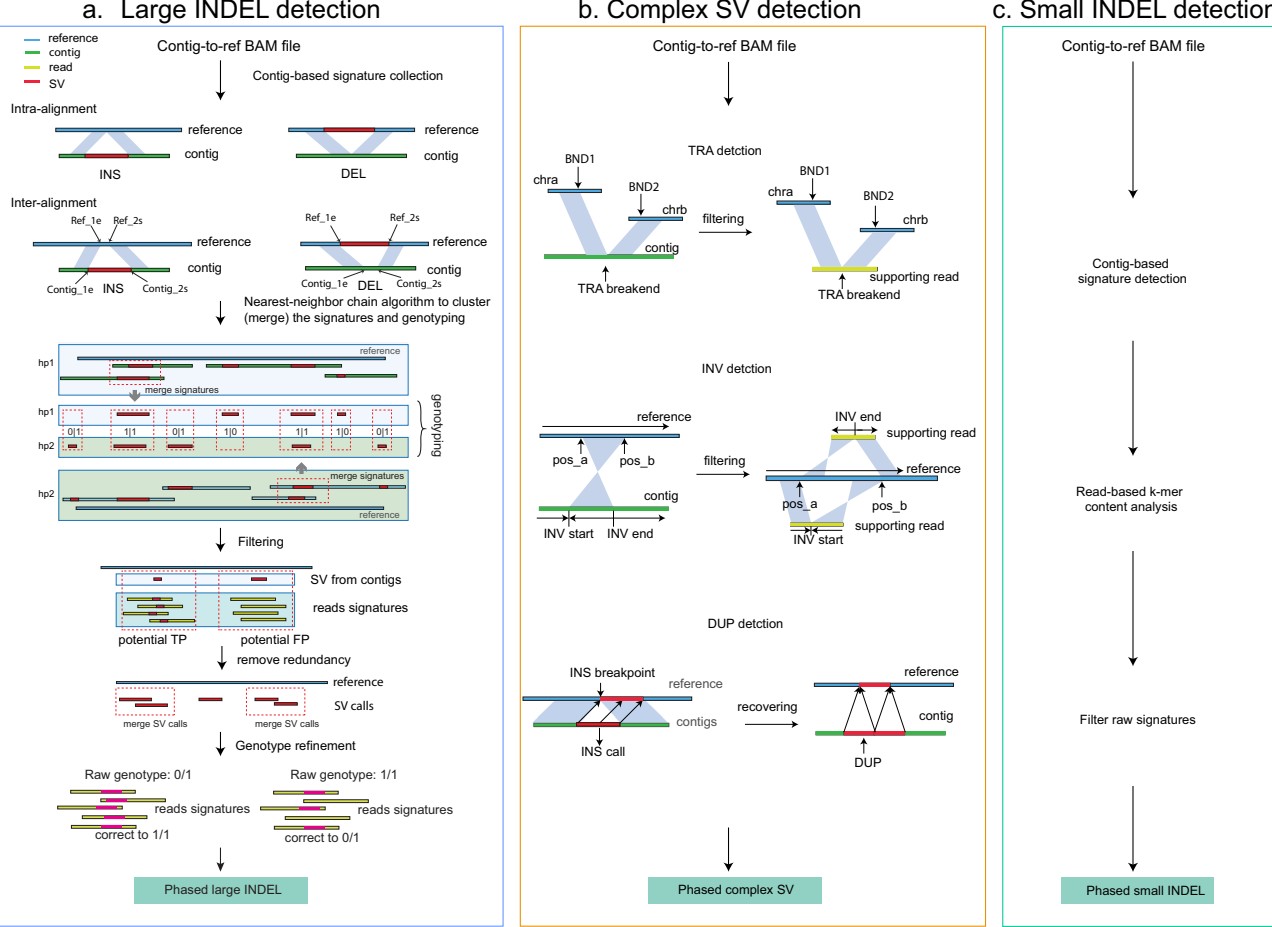

**Fig. 2 | VolcanoSV-vc workflow.** VolcanoSV-vc includes three main modules: **a** large indel SV detection, **b** complex SV detection, and **c** small indel detection. The output of this component is a phased VCF file. Further details are provided in the Methods section.

datasets, VolcanoSV demonstrated consistent and superior performance across different size ranges compared to the other three assembly-based tools (Supplementary Fig. 3).

## VolcanoSV demonstrates proficiency in identifying complex SVs in simulated and real cancer datasets

Up to this point, we have assessed deletion and insertion SVs, which account for most SVs, but other SVs such as translocations (TRA), inversions (INV), and duplications (DUP) also describe different combinations of DNA rearrangements. The lack of benchmarking data for such complex SVs makes it difficult to evaluate tools. To extend the evaluation to complex SV detection, we first applied VolcanoSV and other tools to simulated data. Dipcall was not involved in this analysis since it was not designed to detect complex SVs.

We used VISOR (v1.1.2)[33] to insert SNPs and complex SVs into the hg19 human reference, and then used PBSIM (v3.0.0)[34] to simulate reads for different libraries (Hifi, CLR, and ONT). We compared the SV call results with the initial inserted SVs to calculate F1 and F1 for genotyping accuracy (GT_F1) scores. Detailed simulation and evaluation details are described in the Methods section. Since every reciprocal TRA has four breakends, it is hard to decide whether the genotype is correct or not. We thus only calculated F1 and GT_F1 for INVs and DUPs. In Fig. 4g, VolcanoSV demonstrated the highest F1 performance across all simulated libraries and complex SV types, except for TRAs in Hifi data (Hifi_TRA). Specifically, on CLR and ONT data, VolcanoSV surpassed SVIM-asm, achieving F1 scores that were on average 18%, 23%,

and 63% higher for TRAs, INVs, and DUPs, respectively. Moreover, VolcanoSV's GT_F1 scores were on average 33% and 57% higher than SVIM-asm's GT_F1 for INV and DUP, respectively. For TRAs in Hifi data, VolcanoSV's F1 was only 1% lower than the best F1 by SVIM-asm. PAV was not designed to detect TRAs and DUPs but also had the worst performance on INVs. Noticeably, VolcanoSV achieved much higher F1 and GT_F1 scores for DUPs in all simulated data compared to SVIM-asm, primarily attributed to the effective duplication recovery process implemented in VolcanoSV. With respect to GT_F1, VolcanoSV achieved the best performance on all simulated libraries and complex SV types, except for INVs in Hifi data (Hifi_INV).

We next applied VolcanoSV and SVIM-asm on two publicly available sets of tumor-normal paired libraries (Pacbio CLR and ONT) provided by Talsania et al.[35] and the high confidence HCC1395 somatic SV callset they provided as the benchmark, to further evaluate three classes of somatic complex SVs. To detect somatic SVs, we first applied each assembly-based tool on every library to generate VCF files independently, then used SURVIVOR (v1.0.6)[36] to call somatic variants based on the paired normal-tumor VCFs, and finally compared the somatic SV result to the provided benchmark callset. The high-confidence benchmark callset has a total of 1777 SVs, including 551 insertions, 717 deletions, 146 translocations, 133 inversions, and 230 duplications. Since this high-confidence callset is incomplete, we only plotted recall. VolcanoSV outperformed SVIM-asm in terms of recall across all different libraries and all complex SV types, especially for DUPs (Fig. 4h).

**Table 2 | Performance across 14 datasets for three different types of long-read data**

| INS | | VolcanoSV (2023) | PAV (2021) | SVIM-asm (2020) | Dipcall (2018) | DEL | | VolcanoSV (2023) | PAV (2021) | SVIM-asm (2020) | Dipcall (2018) |
|---|---|---|---|---|---|---|---|---|---|---|---|
| Total benchmark calls (>50): 5281 | | | | | | Total benchmark calls (>50): 4116 | | | | | |
| Hifi_L1 | recall | **94.0%** | 93.11% | 93.09% | 91.35% | Hifi_L1 | recall | **94.63%** | 93.22% | 93.9% | 92.61% |
| | precision | **89.36%** | 86.34% | 86.52% | 72.82% | | precision | **93.72%** | 92.28% | 91.7% | 90.33% |
| | F1 | **91.62%** | 89.6% | 89.68% | 81.03% | | F1 | **94.17%** | 92.75% | 92.79% | 91.46% |
| | gt_concordance | **98.37%** | 95.57% | 97.82% | 77.84% | | gt_concordance | **99.08%** | 97.11% | 97.85% | 96.75% |
| Hifi_L2 | recall | **93.64%** | 93.33% | 92.92% | 91.59% | Hifi_L2 | recall | **94.68%** | 94.12% | 94.56% | 92.76% |
| | precision | **88.1%** | 86.35% | 86.38% | 73.14% | | precision | **93.23%** | 92.04% | 92.01% | 90.11% |
| | F1 | **90.78%** | 89.71% | 89.53% | 81.34% | | F1 | **93.95%** | 93.07% | 93.27% | 91.42% |
| | gt_concordance | **98.02%** | 96.1% | 97.68% | 78.52% | | gt_concordance | **98.9%** | 97.26% | 98.61% | 96.86% |
| Hifi_L3 | recall | **93.86%** | 93.01% | 93.15% | 91.76% | Hifi_L3 | recall | **95.0%** | 93.49% | 94.44% | 92.76% |
| | precision | **87.97%** | 86.28% | 86.36% | 72.97% | | precision | **93.5%** | 91.71% | 91.7% | 89.94% |
| | F1 | **90.82%** | 89.52% | 89.62% | 81.3% | | F1 | **94.24%** | 92.59% | 93.05% | 91.33% |
| | gt_concordance | **98.39%** | 94.89% | 97.95% | 78.29% | | gt_concordance | **99.0%** | 96.62% | 98.59% | 96.83% |
| Hifi_L4 | recall | **93.88%** | 92.99% | 93.18% | 91.5% | Hifi_L4 | recall | **95.07%** | 93.42% | 94.41% | 92.59% |
| | precision | **88.28%** | 86.17% | 86.49% | 72.91% | | precision | **93.57%** | 92.1% | 92.13% | 90.14% |
| | F1 | **91.0%** | 89.45% | 89.71% | 81.16% | | F1 | **94.31%** | 92.75% | 93.26% | 91.35% |
| | gt_concordance | **98.41%** | 95.32% | 97.62% | 78.0% | | gt_concordance | **98.9%** | 96.91% | 98.69% | 96.67% |
| Hifi_L5 | recall | **93.85%** | 93.85% | 93.09% | 91.29% | Hifi_L5 | recall | **94.87%** | 94.53% | 94.53% | 92.42% |
| | precision | **88.22%** | 86.43% | 86.58% | 73.01% | | precision | **93.73%** | 91.99% | 91.92% | 90.04% |
| | F1 | **90.94%** | 89.99% | 89.72% | 81.13% | | F1 | **94.3%** | 93.24% | 93.21% | 91.21% |
| | gt_concordance | **98.39%** | 96.99% | 97.58% | 78.16% | | gt_concordance | **99.15%** | 98.64% | 98.56% | 96.9% |
| Hifi (overall) | recall | **93.85%** | 93.26% | 93.09% | 91.50% | Hifi (overall) | recall | **94.85%** | 93.76% | 94.37% | 92.63% |
| | precision | **88.39%** | 86.31% | 86.47% | 72.97% | | precision | **93.55%** | 92.02% | 91.89% | 90.11% |
| | F1 | **91.03%** | 89.65% | 89.65% | 81.19% | | F1 | **94.19%** | 92.88% | 93.12% | 91.35% |
| | gt_concordance | **98.32%** | 95.77% | 97.73% | 78.16% | | gt_concordance | **99.01%** | 97.31% | 98.46% | 96.80% |
| CLR_L1 | recall | **93.6%** | 91.76% | 93.32% | 88.13% | CLR_L1 | recall | **94.27%** | 89.02% | 91.72% | 86.95% |
| | precision | **86.64%** | 84.15% | 84.62% | 66.76% | | precision | **94.06%** | 91.21% | 92.01% | 88.81% |
| | F1 | **89.99%** | 87.79% | 88.75% | 75.97% | | F1 | **94.16%** | 90.1% | 91.86% | 87.87% |
| | gt_concordance | **98.38%** | 95.87% | 96.77% | 67.6% | | gt_concordance | **98.99%** | 98.39% | 98.46% | 94.36% |
| CLR_L2 | recall | **92.44%** | 91.18% | 89.95% | 20.77% | CLR_L2 | recall | **93.93%** | 86.88% | 86.1% | 21.04% |
| | precision | **86.33%** | 72.05% | 80.21% | 69.61% | | precision | **93.07%** | 81.51% | 88.51% | 82.4% |
| | F1 | **89.28%** | 80.49% | 84.8% | 32.0% | | F1 | **93.49%** | 84.11% | 87.29% | 33.52% |
| | gt_concordance | **96.09%** | 91.75% | 92.0% | 68.0% | | gt_concordance | **98.71%** | 96.0% | 95.23% | 85.45% |
| CLR_L3 | recall | **92.77%** | 91.71% | 90.4% | 47.28% | CLR_L3 | recall | **93.32%** | 84.35% | 84.28% | 46.33% |
| | precision | **87.19%** | 78.91% | 81.51% | 60.4% | | precision | **93.57%** | 88.32% | 90.6% | 83.06% |
| | F1 | **89.89%** | 84.83% | 85.72% | 53.04% | | F1 | **93.44%** | 86.29% | 87.33% | 59.48% |
| | gt_concordance | **96.75%** | 89.24% | 89.86% | 54.27% | | gt_concordance | **98.05%** | 95.65% | 96.17% | 84.27% |
| CLR (overall) | recall | **92.94%** | 91.55% | 91.22% | 52.06% | CLR (overall) | recall | **93.84%** | 86.75% | 87.37% | 51.44% |
| | precision | **86.72%** | 78.37% | 82.11% | 65.59% | | precision | **93.57%** | 87.01% | 90.37% | 84.76% |
| | F1 | **89.72%** | 84.37% | 86.42% | 53.67% | | F1 | **93.70%** | 86.83% | 88.83% | 60.29% |
| | gt_concordance | **97.07%** | 92.29% | 92.88% | 63.29% | | gt_concordance | **98.58%** | 96.68% | 96.62% | 88.03% |
| ONT_L1 | recall | 93.47% | 91.42% | **93.73%** | 90.32% | ONT_L1 | recall | **94.46%** | 91.28% | 94.34% | 91.67% |
| | precision | **87.66%** | 85.3% | 85.57% | 65.49% | | precision | **93.55%** | 91.34% | 91.62% | 86.98% |
| | F1 | **90.47%** | 88.26% | 89.46% | 75.93% | | F1 | **94.0%** | 91.31% | 92.96% | 89.26% |
| | gt_concordance | **98.42%** | 97.33% | 98.08% | 63.71% | | gt_concordance | **99.18%** | 98.72% | 98.79% | 93.32% |
| ONT_L2 | recall | 91.93% | **92.96%** | 92.54% | 84.79% | ONT_L2 | recall | 92.61% | **94.07%** | 93.83% | 88.82% |
| | precision | **88.39%** | 84.74% | 84.58% | 64.27% | | precision | **91.72%** | 90.13% | 90.07% | 83.68% |
| | F1 | **90.12%** | 88.66% | 88.38% | 73.12% | | F1 | **92.17%** | 92.06% | 91.91% | 86.18% |
| | gt_concordance | 97.67% | 97.23% | **97.83%** | 59.65% | | gt_concordance | **99.11%** | 98.42% | 98.42% | 89.36% |
| ONT_L3 | recall | **93.51%** | 92.71% | 93.33% | 89.45% | ONT_L3 | recall | **93.95%** | 92.78% | 93.9% | 90.72% |
| | precision | **87.32%** | 86.03% | 85.28% | 65.47% | | precision | **92.82%** | 91.1% | 90.86% | 86.48% |
| | F1 | **90.31%** | 89.25% | 89.12% | 75.6% | | F1 | **93.38%** | 91.94% | 92.35% | 88.55% |

**Table 2 (continued) | Performance across 14 datasets for three different types of long-read data**

| INS | | VolcanoSV (2023) | PAV (2021) | SVIM-asm (2020) | Dipcall (2018) | DEL | | VolcanoSV (2023) | PAV (2021) | SVIM-asm (2020) | Dipcall (2018) |
|---|---|---|---|---|---|---|---|---|---|---|---|
| Total benchmark calls (>50): 5281 | | | | | | Total benchmark calls (>50): 4116 | | | | | |
| | gt_concordance | **98.22%** | 97.22% | 98.19% | 63.74% | | gt_concordance | **99.02%** | 98.74% | 98.78% | 92.64% |
| ONT_L4 | recall | **92.37%** | 91.65% | 91.1% | 73.49% | ONT_L4 | recall | **92.69%** | 92.03% | 92.35% | 76.21% |
| | precision | **85.88%** | 82.79% | 82.86% | 62.55% | | precision | **92.35%** | 89.7% | 90.44% | 82.99% |
| | F1 | **89.01%** | 87.0% | 86.79% | 67.58% | | F1 | **92.52%** | 90.85% | 91.38% | 79.46% |
| | gt_concordance | **97.31%** | 95.81% | 96.57% | 55.45% | | gt_concordance | **98.87%** | 97.1% | 97.08% | 86.01% |
| ONT_L5 | recall | **93.13%** | 89.24% | 92.9% | 86.8% | ONT_L5 | recall | 93.59% | 89.04% | **93.71%** | 88.61% |
| | precision | **87.88%** | 84.72% | 85.06% | 63.83% | | precision | **92.82%** | 90.88% | 91.05% | 84.66% |
| | F1 | **90.43%** | 86.92% | 88.8% | 73.56% | | F1 | **93.2%** | 89.95% | 92.36% | 86.59% |
| | gt_concordance | **97.93%** | 97.18% | 97.92% | 59.25% | | gt_concordance | **99.09%** | 98.72% | 98.73% | 90.18% |
| ONT_L6 | recall | 92.65% | 90.72% | **92.67%** | 86.99% | ONT_L6 | recall | **93.73%** | 90.89% | 93.71% | 89.14% |
| | precision | **87.97%** | 84.78% | 84.48% | 64.42% | | precision | **93.3%** | 90.96% | 91.03% | 85.39% |
| | F1 | **90.25%** | 87.65% | 88.39% | 74.03% | | F1 | **93.52%** | 90.92% | 92.35% | 87.22% |
| | gt_concordance | **98.43%** | 97.1% | 98.1% | 60.58% | | gt_concordance | **99.07%** | 98.74% | 98.81% | 91.06% |
| ONT (overall) | recall | **92.84%** | 91.45% | 92.71% | 85.31% | ONT (overall) | recall | 93.50% | 91.68% | **93.64%** | 87.53% |
| | precision | **87.52%** | 84.73% | 84.64% | 64.34% | | precision | **92.76%** | 90.69% | 90.84% | 85.03% |
| | F1 | **90.10%** | 87.96% | 88.49% | 73.30% | | F1 | **93.13%** | 91.17% | 92.22% | 86.21% |
| | gt_concordance | **98.00%** | 96.98% | 97.78% | 60.40% | | gt_concordance | **99.06%** | 98.41% | 98.44% | 90.43% |

The table lists recall, precision, F1 (%), and genotype accuracy (represented by gt_concordance). The highest recall, precision, F1, and genotype accuracy scores among four benchmarked SV callers are marked in bold. For each PacBio HiFi, CLR, and ONT long read type, the overall average performance is also shown. The following parameter setting was used in Truvari for benchmarking: $p = 0.5$, $P = 0.5$, $r = 500$, $O = 0.01$. Source data are provided as a Source Data file.

Specifically, on PacBio CLR data, VolcanoSV's recall was 23%, 18%, and 12% higher than SVIM-asm for TRAs, INVs, and DUPs respectively. On ONT data, VolcanoSV's recall was 13%, 9%, and 17% higher than SVIM-asm for TRAs, INVs, and DUPs respectively. Although VolcanoSV outperformed SVIM-asm in detecting complex SVs, its recall was relatively lower compared to its performance for simulated data or indel SVs. To understand the reason behind this low recall and whether VolcanoSV predominantly detects common complex SVs identifiable by other long-read datasets, we analyzed the overlapping VolcanoSV calls between paired normal and tumor HCC1395 samples, and HG002. Detailed results and discussion are provided in Supplementary Fig. 4 and Supplementary Notes 1.1. Our analysis revealed that 1) VolcanoSV failed to detect a sufficient number of unique somatic complex SVs in the tumor sample, in addition to the germline SVs; 2) The high-confidence benchmark callset included SV calls only from alignment-based tools, and therefore using it to evaluate VolcanoSV might introduce bias. Overall, it is still challenging to solely use assembled contigs to detect complex SVs due to the limitations of assembly algorithms and the complexity of graph construction. Many genome assembly algorithms build contigs by following the simplest paths through overlapping reads, which may miss complex SVs. These variants create irregular patterns that do not fit into the straightforward paths the algorithms usually prefer or disrupt the continuity of the graph.

### VolcanoSV maintained superior performance on low coverage datasets

To further benchmark VolcanoSV's robustness to different sequencing coverages against the three assembly-based SV callers (PAV, SVIM-asm, and Dipcall), we evaluated SV calling results on subsampled Hifi_L1, CLR_L1, and ONT_L1. All three datasets were subsampled to 40x, 30x, 20x, 10x, and 5x coverage using rasusa (v0.6.0)[37]. Hifi_L1 and CLR_L1 were additionally subsampled to 50x coverage due to their higher original coverage.

The four tools exhibited similar subsampling effects on Hifi_L1 (Fig. 5a and Supplementary Table 6). Noticeable changes in terms of recall, precision, and F1 were only observed after the coverage dropped to 5x. For both insertions and deletions, VolcanoSV showed greater robustness to subsampling effects on Hifi_L1 compared to the other three tools, since it could still preserve high F1 scores at 10x coverage. At 5x coverage, SVIM-asm kept the highest F1, followed by VolcanoSV.

Two main patterns were observed in the subsampling effects on CLR_L1 and ONT_L1 (Fig. 5b, c, Supplementary Table 7, and Supplementary Table 8). VolcanoSV and Dipcall showed relatively stable precision across different coverages, but lower recall at lower coverages (5–10×). On the other hand, PAV and SVIM-asm were able to maintain relatively better recall as coverage decreased, however, their precision declined quickly as coverage dropped to lower coverages. Considering F1 as the metric, VolcanoSV still demonstrated greater robustness against subsampling on CLR_L1 and ONT_L1, as it exhibited better F1 scores compared to the other three tools when the coverage dropped to 10× and 5×.

We also examined subsampling effects on genotyping accuracy. The genotyping performance shared a similar trend with the overall accuracy for the subsampling effects. Across all coverages (5–50×), VolcanoSV maintained the best genotype accuracy compared to the other three tools.

### VolcanoSV is robust to SV evaluation parameters

Due to the complex nature of structural variants, SV benchmark tools such as Truvari usually do not require a detected SV to have exactly the same breakpoints and sequence as the true SV to be considered correct, but instead use a set of evaluation parameters to control the matching tolerance. While the matching tolerance can facilitate fair SV comparisons, the choice of the evaluation parameters is usually empirical and subjective, which increases the uncertainty of the comparison outcomes and could limit our understanding of the SV calling performance. A lenient threshold might underestimate

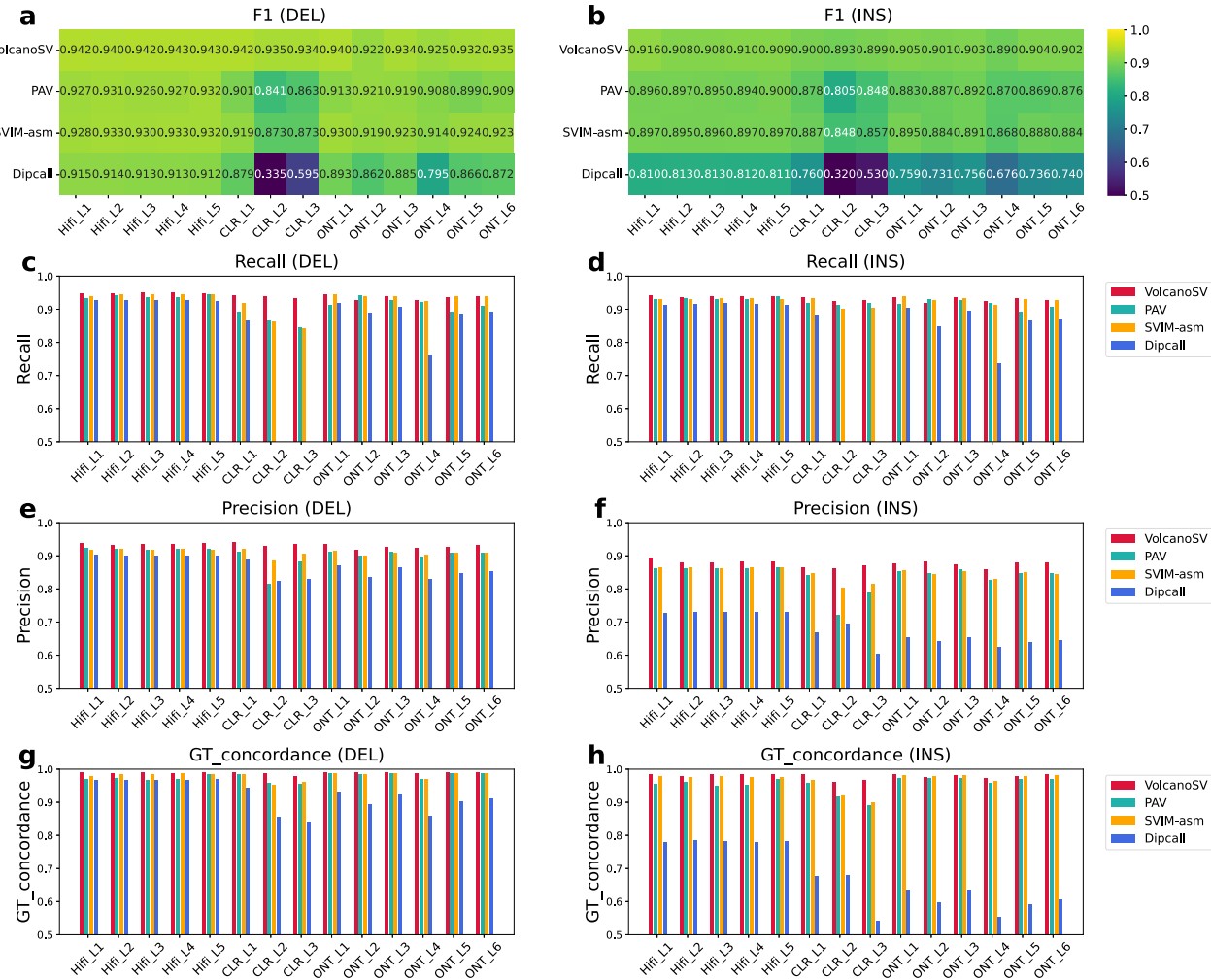

**Fig. 3 | Cross datasets evaluation against GIAB HOO2 benchmark. a**, **b** F1 heatmap for deletions (DEL) and insertions (INS) by four assembly-based tools. **c**, **d** Recall bar plots for insertions deletion (DEL) and insertions (INS) by four assembly-based tools. **e**, **f** Precision bar plots for insertions deletion (DEL) and insertions (INS) by four assembly-based tools. **g**, **h** Genotype accuracy (represented by GT_concordance) bar plots for insertions deletion (DEL) and insertions (INS) by four assembly-based tools. Source data are provided as a Source Data file.

distinctions among SV callers, while a stringent threshold could exaggerate the superiority of a particular SV caller. Therefore, we used a grid search experiment to explore Truvari's parameters to thoroughly investigate the robustness of VolcanoSV and other assembly-based tools[38]. As described before, evaluation parameters include pctstim ($p$), pctsize ($P$), pctovl ($O$), and refdist ($r$). In general, higher values of $p$, $P$, and $O$, along with lower values of $r$, establish more stringent comparison criteria. This means that the SVs being compared will need to exhibit greater sequence similarity, allele size similarity, reciprocal overlap ratio, or closer proximity to the reference sequence in order to be classified as the same SV. More descriptions of Truvari and its parameters are provided in the Methods section. Specifically, in our grid search SV evaluation experiments, we varied $p$, $P$, $O$ from 0 to 1 in increments of 0.1, and $r$ from 0 to 1 kb in increments of 100 bp.

We first evaluated deletions on Hifi_L1. Parameters $p$ and $O$ had the most significant effects. As illustrated in Fig. 6a, when $p$ and $O$ increased (i.e. a more stringent correspondence was required between the call and the gold standard to be accepted as a true positive), all four assembly-based tools, Dipcall, SVIM-asm, PAV, and VolcanoSV demonstrated stable and high performance across the grid search and could still maintain reasonable F1 scores under the most stringent parameter settings. Among them, VolcanoSV showed the most robust

performance with the changes of evaluation parameters and maintained the highest F1 score across all entries. Parameters $P$ and $r$ had little effect on deletion evaluation for all methods unless they were set to 1.0 or 0 bp, respectively.

In terms of evaluating insertions on Hifi_L1, parameters $p$ and $r$ were chosen as the representative pair to compare performance across methods. Compared to deletions, insertions called by assembly-based tools were slightly more robust to the changes of evaluation parameters (Fig. 6d), except for the most stringent conditions ($p = 1.0$ or $r = 0$). SVIM-asm, PAV, and VolcanoSV, maintained relatively higher performance across grid searches than Dipcall. Overall, VolcanoSV achieved higher F1 than SVIM-asm and PAV for each grid. The parameter $O$ was not used since it is not applicable to insertion evaluation. The parameter $P$ only had a significant effect on insertion evaluation for SV callers when it was set to 1.0. This result was consistent with its effect on deletion evaluation. We have also observed similar patterns on grid searches for all methods on CLR_L1 and ONT_L1 (Supplementary Figs. 5 and 6).

To reveal why assembly-based methods like VolcanoSV were robust to changes of evaluation parameters, we further analyzed the distribution of SV breakpoint and alternate allele sequence similarity for several representative tools. The breakpoint shift was determined by calculating the maximum difference in reference locations between

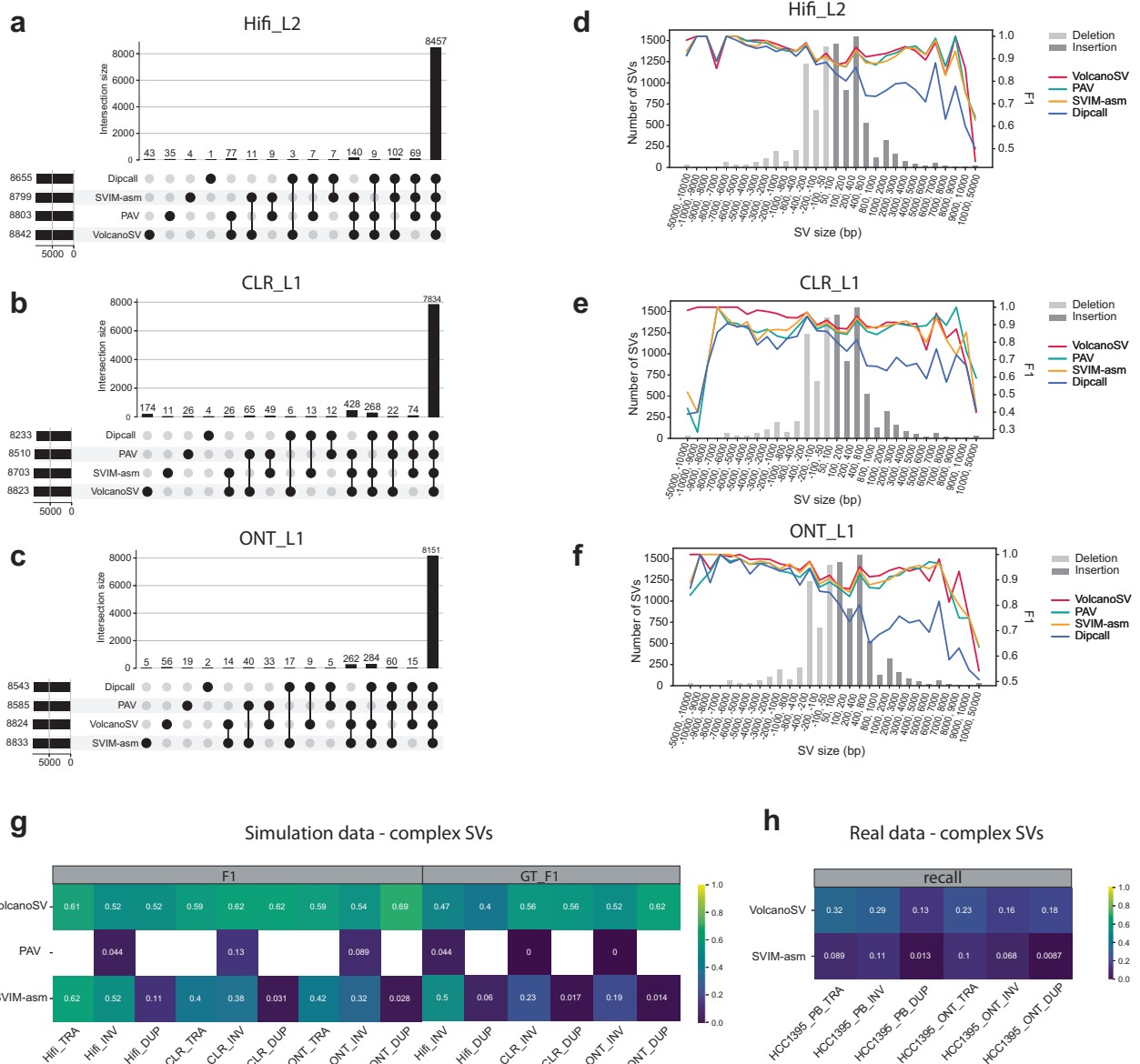

**Fig. 4 | Overlapping calls, size distribution, and accuracy for SV discovery and complex SV analysis. a–c** UpSet plot for analysis of shared and unique true positive (TP) calls between different assembly-based tools. **d–f** F1 accuracy of SV detection at different size ranges. The negative size range represents deletions and the positive size range represents insertions. The bar plot shows benchmark SV distribution at different size ranges. The line plot shows the F1 score of four different methods. **g** F1 and GT_F1 heatmap for complex SV detection on simulated data. **h** The recall heatmap for complex somatic SV detection on real data. Source data are provided as a Source Data file.

the two compared SV calls. The similarity in SV sequences was computed based on the edit distance between the two compared SV calls. The results showed that all four tools had a near zero breakpoint shift and near 100% SV sequence similarity with the benchmark callset (Fig. 6b, c, e, f). This result underscores that the ability to accurately capture SV breakpoints and alternative allele sequences contributes to the tool's resilience in rigorous SV evaluation scenarios, a capability possessed by all assembly-based methods.

**VolcanoSV-vc attains low false discovery rate**
We employed the complete VolcanoSV pipeline to produce diploid assembly, ultimately facilitating the detection of all types of structural variants. VolcanoSV-vc, serving as the assembly-based SV calling component of VolcanoSV, is versatile enough to function as a standalone tool, accepting assembly inputs from other assemblers. We thus investigated the SV calling performance of VolcanoSV-vc by

taking hifiasm's assembly as input for Hifi data, and Flye + HapDup's assembly as input for CLR and ONT data. In comparison to the three other assembly-based methods (PAV, SVIM-asm, and Dipcall), VolcanoSV-vc achieved the best F1 across all 14 PacBio Hifi, CLR, and ONT datasets (Supplementary Tables 9–11 and Supplementary Fig. 7).

Although benchmarking against the GIAB SV gold standard is an efficient and precise procedure to evaluate and compare the SV calling performance of different tools, the GIAB gold standard SV callset is not a complete set and could also contain false positives. We thus used T2T-CHM13 (v2.0) as an additional reference genome and the long reads from the CHM13 sample to call SVs and evaluate the false discovery rate. Specifically, we used hifiasm (v0.16) to assemble long reads into dual contigs, and applied different assembly-based SV callers to the contigs against both T2T-CHM13 and GRCh38 reference genomes (v2.1.0). We did not use the VolcanoSV-asm component for the diploid assembly since the CHM13 sample derives from a single

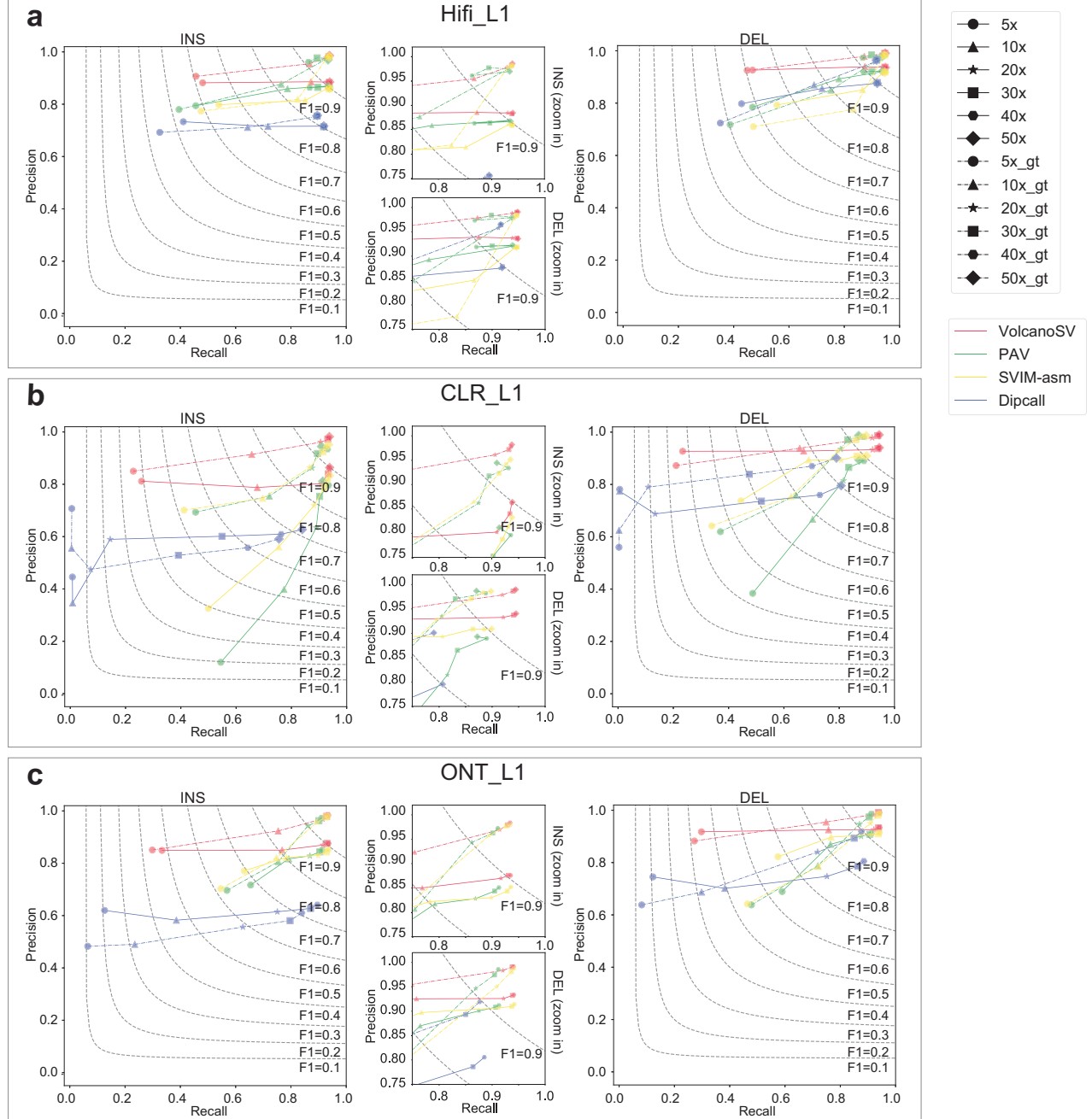

**Fig. 5 | Subsampling effect for different methods. a** Recall-precision-F1 curves show the subsampling effect on deletion and insertion by different tools on Hifi_L1. **b** Recall-precision-F1 curves show the subsampling effect on deletion and insertion by different tools on CLR_L1. **c** Recall-precision-F1 curves show the subsampling effect on deletion and insertion by different tools on ONT_L1. The coverage depth varies from 5×, 10×, 20×, 30×, 40× to 50×. Solid lines with markers are for different coverage depths, and corresponding dashed lines are for genotyping (gt) accuracy. For both insertions and deletions, we zoom in on the top right part of the plot to demonstrate the curves more clearly. Source data are provided as a Source Data file.

homozygous complete hydatidiform mole and is not suitable for the haplotype-phasing module to achieve diploid assembly. The false discovery rate was computed as the ratio of the number of detected SVs against CHM13 to the number of SVs against GRCh38.

Given the remarkably precise T2T CHM13 assembly generated with multiple complementary technologies, ideally, no SV should be discovered when the contigs assembled from CHM13 long reads are aligned to the T2T-CHM13 reference. However, we did observe a few SV calls when performing the variant calling due to the imperfection of assembly and bias introduced by the aligner and variant caller. Therefore, the number of false discoveries can serve as an objective measure of how robust a variant caller is. We performed this experiment on two Hifi datasets from CHM13 (denoted as Hifi_L6 and Hifi_L7) for different assembly-based SV callers, and the results were collected in Table 3. We observed that overall, VolcanoSV achieved the lowest false discovery rate (FDR) compared with the other three tools. Specifically, on Hifi_L6, VolcanoSV achieved the lowest baseline FDR at 3.03%, followed by Dipcall (3.05%), SVIM-asm (4.33%), and PAV (9.21%).

In terms of total calls against the T2T-CHM13 reference (relative to which the false discovery rate is calculated), VolcanoSV ranked second with 902 calls, following behind Dipcall, which had 737 calls. In terms of total calls against the GRCh38 reference, VolcanoSV had the highest

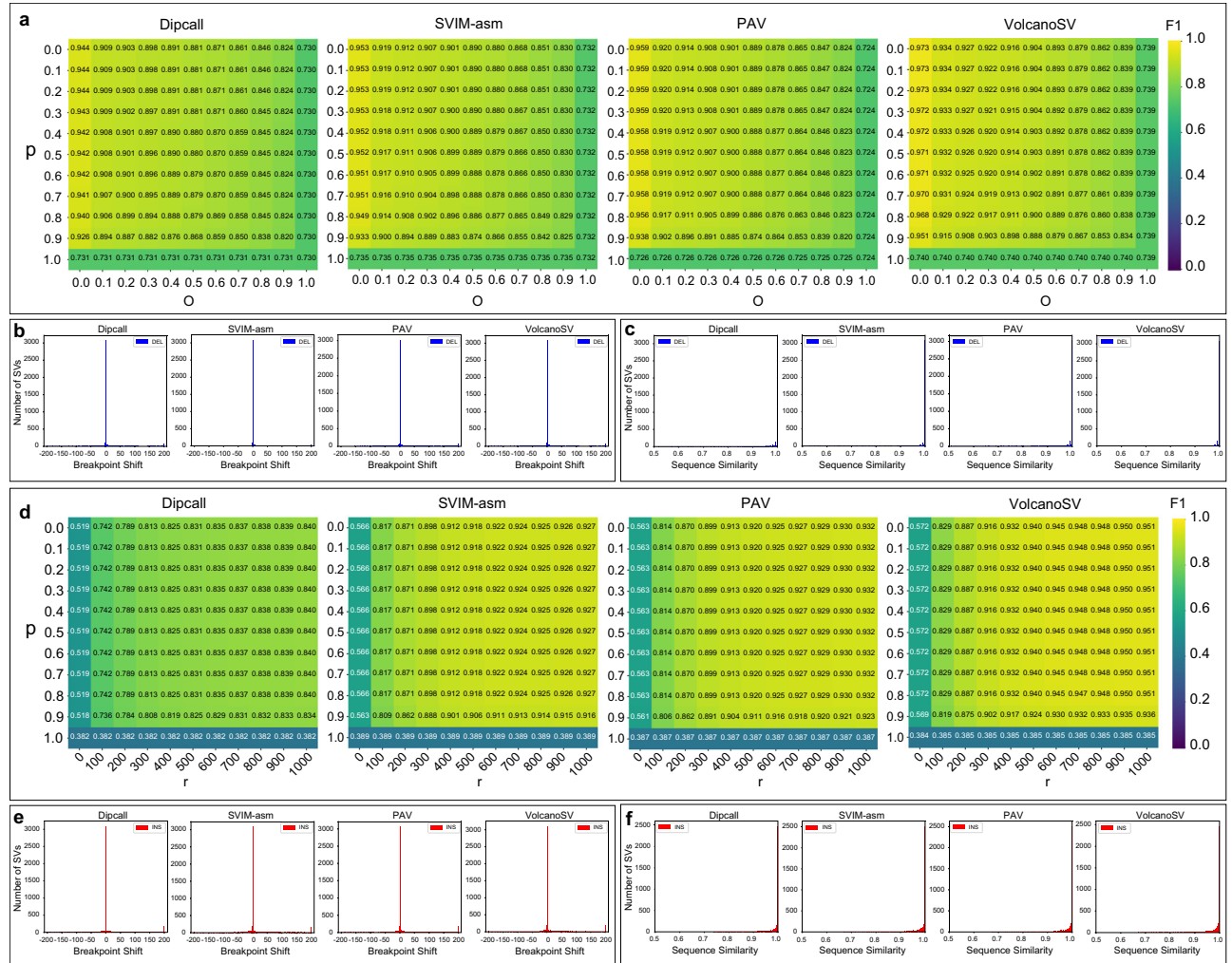

**Fig. 6 | F1 accuracy by tuning different evaluation parameters and distribution of breakpoint shift and alternate allele sequence similarity for SVs on Hifi_L1. a** Grid search heatmap of F1 values for deletions by different assembly-based tools. **b** Distribution of breakpoint shift for deletions by assembly-based tools. **c** Distribution of alternate sequence similarity for deletions by assembly-based tools. **d–f** Equivalent visual representations as shown in **a–c** for insertions. Source data are provided as a Source Data file.

number (29748), followed by SVIM-asm (26384), Dipcall (24187), and PAV (17540). On Hifi_L7, VolcanoSV maintained its lead with the lowest baseline FDR at 2.22%, followed by Dipcall (2.94%), SVIM-asm (3.36%),

### Table 3 | Comparison of false discovery rate

| Library | Tool | GRCh38 calls | T2T calls (FDs) | Baseline FDR | % T2T calls (FDs) in heterochromatin regions |
|---|---|---|---|---|---|
| Hifi_L6 | VolcanoSV | 29748 | 902 | **3.03%** | 92.7% |
| | PAV | 17540 | 1616 | 9.21% | 96.4% |
| | SVIM-asm | 26384 | 1143 | 4.33% | 94.2% |
| | Dipcall | 24187 | 737 | 3.05% | 92.7% |
| Hifi_L7 | VolcanoSV | 26737 | 593 | **2.22%** | 79.9% |
| | PAV | 17563 | 794 | 4.52% | 85.8% |
| | SVIM-asm | 25480 | 857 | 3.36% | 81.8% |
| | Dipcall | 23976 | 705 | 2.94% | 82.4% |

The number of SV calls against both the reference genome GRCh38 and the T2T-CHM13 assembly are included. The false discovery rate (FDR) is calculated as the ratio of the number of SVs against T2T-CHM13 to the number of SVs against GRCh38 for each tool. The lowest FDR is highlighted in bold for each dataset. "% T2T calls (FDs) in heterochromatin regions" denotes the percentage of T2T calls (FDs) by each tool in heterochromatin regions. "FDs" stands for false discoveries. Source data are provided as a Source Data file.

and PAV (4.52%). For total calls against the T2T-CHM13 reference, VolcanoSV had the lowest false calls (593), followed by Dipcall (705), PAV (794), and SVIM-asm (857). In terms of total calls against the GRCh38 reference, VolcanoSV had the highest number (26737), followed by SVIM-asm (25480), Dipcall (23976), and PAV (17563).

We conducted additional analysis on the falsely discovered SVs based on the T2T-CHM13 reference. The results, illustrated in the UCSC Genome Browser (Supplementary Figs. 8 and 9) and Table 3, showed that 92.6%, 96.4%, 94.2%, and 92.7% of FD for Hifi_L6 by VolcanoSV, PAV, SVIM-asm, and Dipcall, respectively, were located in heterochromatin regions like centromeres and telomeres. For Hifi_L7, the percentages were lower, with 79.9%, 85.8%, 81.8%, and 82.4% of SVs identified by VolcanoSV, PAV, SVIM-asm, and Dipcall, respectively, also located in these complex regions. Notably, the majority of FDs were concentrated in chromosomes 1, 7, 9, 15, and 16. Heterochromatin regions pose challenges for assembly, often requiring specialized graph-based methods for accurate resolution.

### Phased SNPs, small indels and SVs

While VolcanoSV was originally designed with a focus on detecting structural variants (SVs), the diploid assembly also provides the potential for uncovering small variants (≤50 bp) across the entire genome. We benchmarked SNPs and small indels by VolcanoSV with

PAV and Dipcall since they both detect small variants through the assembly. In Hifi data (Supplementary Table 12), VolcanoSV demonstrated the highest F1 scores in SNPs for four out of five datasets and excelled in small indels for all five datasets, achieving peak F1 scores of 99.42% for SNPs and 98.34% for small indels. When applied to CLR data (Supplementary Table 13), VolcanoSV outperformed PAV and Dipcall, achieving the best F1 scores in both SNPs and small indels across all three datasets. Similarly, for ONT data (Supplementary Table 14), VolcanoSV exhibited superior performance in SNPs and small indels for all three datasets, although all three tools displayed low precision in small indel calls. It is noteworthy that detecting small indels based on assembly proved to be ineffective for ONT data.

We note that VolcanoSV includes a haplotype phasing module, enabling not only detection but also phasing of all types of variants, including SNPs, small indels, and SVs. The end result is the production of a phased VCF file.

## Computational costs

To assess the runtime and memory usage of all tools, we used three different long-read datasets as representatives. All tools were tested on AMD EPYC 7452 Processor CPUs with 32 cores and 1TB memory. We first investigated the computation cost for the assembly phase. For Hifi_L1 (Supplementary Table 15), VolcanoSV-asm and hifiasm finished within 1474 and 440 CPU hours with peak memory usage reaching around 168GB and 214GB, respectively. In the CLR_L1 library, VolcanoSV-asm and Fly + HapDup finished within 2582 and 10673 CPU hours with peak memory usage of 259GB and 691GB, respectively; Finally, for ONT_L1, VolcanoSV-asm and Fly + HapDup finished in 4397 and 7886 CPU hours with peak memory usage of 85GB and 337GB, respectively. In summary, owing to the phasing and local assembly modules, VolcanoSV-asm demonstrated lower memory consumption and runtime than Fly + HapDup; it required more runtime and less memory compared to hifiasm. However, assembly procedures for assembly-based tools are often more computationally expensive than alignment procedures for alignment-based tools. For instance, when considering the same three libraries, alignment procedures utilizing minimap2 (v2.24-r1122)[39] only consume 189, 118, and 116 CPU hours, with peak memory usage reaching approximately 31GB, 29GB, and 31GB, respectively.

Secondly, we benchmarked the computational costs for the assembly-based variant calling phase. For Hifi_L1 (Supplementary Table 16), VolcanoSV-vc, PAV, SVIM-asm, and Dipcall finished within 21, 97, 17, and 11 CPU hours with peak memory of 21GB, 101GB, 37GB, and 48GB, respectively. For CLR_L1, VolcanoSV-vc, PAV, SVIM-asm, and Dipcall finished within 123, 1098, 7, and 10 CPU hours with peak memory of 215GB, 67GB, 26GB, and 42GB, respectively, For ONT_L1, VolcanoSV-vc, PAV, SVIM-asm, and Dipcall finished within 32, 133, 8, and 8 CPU hours with peak memory of 34GB, 53GB, 34GB, and 47GB, respectively. In summary, due to the variant filtering and refinement module, VolcanoSV-vc consumed less memory but more runtime than SVIM-asm and Dipcall on Hifi and ONT data. However, on CLR data, VolcanoSV-vc exhibited a substantial memory consumption. PAV consistently required significantly more runtime than other assembly-based tools. Compared to state-of-the-art alignment-based tools[17–19,38], VolcanoSV-vc shows no significant difference in computation cost.

## Selecting assemblers for regions enriched in segmental duplications

We selected hifiasm and Flye as the default assemblers for various long-read datasets after evaluating several upstream long-read assemblers, including hifiasm[26], Flye[27], Peregrine[40], wtdbg2[41], IPA[42], HiCanu[43], and Shasta[44]. Our assessment considered their impact on VolcanoSV's downstream large indel SV calling performance. These two assemblers consistently demonstrated superior performance on PacBio Hifi (hifiasm), and for CLR and ONT datasets (Flye),

respectively. However, users also have the flexibility within the assembly module in VolcanoSV to choose the most appropriate or new assemblers to fulfill their specific requirements. For example, when assembling regions enriched in segmental duplications (SDs), these two assemblers may not be the most suitable choice. To investigate the performance of different assemblers in SD-enriched regions, we employed Flagger (v0.3.3)[45] to detect misassemblies enriched in SDs in three representative libraries. Flagger is a read-based pipeline that maps long reads to the phased diploid assembly in a haplotype-aware manner. It identifies coverage inconsistencies within these read mappings that are likely due to assembly errors. Details are provided in the Methods section.

For ONT_L1, we additionally employed wtdbg2 (v2.2.5), Shasta (v0.10.0), and NextDenovo (v2.5.2)[46] in VolcanoSV to perform local assembly for all phase blocks. Using Flagger, we first identified collapsed components in diploid assembly by VolcanoSV using different assemblers. These collapsed components represent regions with additional copies in the underlying genome that have been collapsed into a single copy, indicating potential misassemblies for SDs. We then further assessed the SD reliability of the Flagger annotation for collapsed components. To achieve this, we aligned the collapsed components from all diploid assemblies to the GRCh38 reference genome and intersected them with the SD annotations for HG002 based on GRCh38. We then calculated and compared the total length of SD annotation regions that overlap with collapsed regions by Flagger across all assemblies. The results demonstrated that VolcanoSV assembly using Flye, wtdbg2, Shasta, and NextDenovo generated 34.9 MB, 47.6 MB, 34.2 MB, and 22.7 MB of collapsed components, respectively, which suggested collapsed SDs.

For CLR_L1, we additionally employed wtdbg2, and NextDenovo in VolcanoSV to perform local assembly for all phase blocks. The results demonstrated that VolcanoSV assembly using Flye, wtdbg2, and NextDenovo generated 28.8 MB, 32 MB, and 30.2 MB of collapsed components, respectively, which suggested collapsed SDs. For Hifi_L1, we additionally employed Hicanu (v2.1.1) in VolcanoSV to perform local assembly for all phase blocks. The results demonstrated that VolcanoSV assembly using hifiasm and Hicanu generated 17.4 MB and 15.7 MB of collapsed components, respectively, which suggested collapsed SDs.

These results suggested that NextDenovo in VolcanoSV assembled fewer collapsed regions than Flye in ONT_L1, indicating its potential superiority for regions enriched in SDs. Similarly, Hicanu in VolcanoSV outperformed hifiasm in Hifi_L1. Notably, Flye in VolcanoSV assembled the fewest collapsed regions in CLR_L1 compared to other assemblers.

## Discussion

In this work, we introduce VolcanoSV, a reference-assisted, haplotype-resolved, assembly-based, structural variant detection method. It relies on a sophisticated *k*-mer-based reads partitioning method, and performs contig-based reads signature collection and rigorous FP filtering, followed by genotype correction. The combination of these steps enables VolcanoSV to attain remarkable performance in structural variation discovery. Our benchmarking analysis, evaluated against the ground-truth HG002 SV callset from genome in a bottle (GIAB), demonstrated superior levels of recall, precision, F1 score, and genotype accuracy compared to existing state-of-the-art assembly-based tools across a diverse range of datasets, including low-coverage (10×) ones. Moreover, VolcanoSV was robust across SV evaluation parameter settings and achieved accurate breakpoints and novel sequence identification. It consistently achieved the highest F1 score across all combinations of evaluation parameters compared to other assembly-based tools. Due to the limited availability of ground-truth SV calls other than for large insertions and deletions, VolcanoSV was benchmarked for duplications, inversions, and translocations in both

simulated and real cancer datasets. This analysis demonstrated a high level of performance in complex SV detection. Furthermore, because the ground-truth SV callset is incomplete, VolcanoSV-vc was further benchmarked to detect SVs from the CHM13 sample by employing the new reference genome T2T-CHM13. VolcanoSV-vc showed the lowest false discovery rate of any available tool. However, because the diploid assembly key component "VolcanoSV-asm" of VolcanoSV aims to achieve haplotype-resolved diploid assembly via heterozygous SNPs, it is not suitable for samples like CHM13, which is derived from a single homozygous complete hydatidiform mole to perform diploid assembly. In addition to SV detection, VolcanoSV also demonstrated superior performance in small variants detection and assembled a phased VCF file as the final product.

In comparison to existing SV callers, VolcanoSV is a comprehensive workflow, and we recommend users to use it directly to generate diploid assembly and detect variants. However, the VolcanoSV-asm component can be used independently to generate diploid assembly inputs for other assembly-based tools. Additionally, the VolcanoSV-vc component can be independently used to detect variants by taking any diploid assembly as input from other assembly tools. We highlight two noteworthy and advantageous features of VolcanoSV: (1) It combines both the contig-based signatures and read-based signatures to jointly infer and filter structural variants, resulting in a superior level of sensitivity and precision. Alignment-based methods heavily rely on the reads' quality and the aligner's capabilities. In regions with high repetition or other complexities, most aligners struggle to confidently map reads, leading to reduced sensitivity in SV detection. Our reference-assisted de novo assembly-based approach generates longer contigs, effectively resolving the issue of alignment ambiguity. However, genome-wide assembly may introduce artifacts and produce false calls. Therefore, employing contig-based signatures to generate a draft SV map and combining it with read-based signatures for false positive filtering represents a judicious hybrid approach, harmonizing the strengths of both approaches and compensating for their respective limitations. (2) VolcanoSV implements a *k*-mer-based reads partitioning method, significantly improving the ratio of reads that are suitable for diploid assembly, therefore boosting the variant calling performance. We consider this *k*-mer-based method as an innovative and complementary strategy for phasing. Current phasing software can only make use of the heterozygous SNP information, while the heterozygosity introduced by indels or SVs is ignored. This *k*-mer-based method takes into consideration the heterozygosity caused by indel and any other variants, therefore facilitating the generation of a more comprehensive haplotype-resolved assembly. Although assembly-based variant detection approaches are often much more demanding in computational resources than alignment-based approaches, assembly-based approaches are more likely to generate comprehensive SV calls with precise breakpoints and alternate sequences across the whole genome. VolcanoSV incorporates hifiasm and Flye as default assemblers for local assembly, however, users have the flexibility to utilize an appropriate or new assembler based on their needs.

## Methods

### VolcanoSV workflow

VolcanoSV uses a reference sequence and long reads data to generate a high-quality haplotype-resolved diploid assembly, from which it then comprehensively detects SVs and removes false positives. Not only limited to SVs, VolcanoSV also integrates two modules to collect SNPs and small indels, and further refine these small variants. For the final product, VolcanoSV provides users with a complete map of all types of phased variants. The main workflow of VolcanoSV consists of two key components (VolcanoSV-asm and VolcanoSV-vc) interconnected through six conceptual modules: (a) Partitioning reads relying on haplotype phasing (VolcanoSV-asm, Fig. 1); (b) Unphased reads assignment through unique *k*-mer similarity analysis (VolcanoSV-asm,

Fig. 1); (c) Haplotype-aware local assembly via augmented phase blocks (VolcanoSV-asm, Fig. 1); (d) Contig alignment-based large indel SV detection and refinement (VolcanoSV-vc, Fig. 2); (e) Complex SV collection, recovery and filtering (VolcanoSV-vc, Fig. 2); (f) Small indel collection and refinement (VolcanoSV-vc, Fig. 2). The final output is a phased Variant Call Format (VCF) file. Details for each module are described in the following sections.

**Partitioning reads relying on haplotype phasing.** VolcanoSV is a reference-assisted haplotype-resolved assembly approach using as input a phased alignment BAM file and a phased VCF file including all heterozygous SNVs. For haplotype phasing, VolcanoSV integrates Longshot (v0.4.1), a recently developed tool for SNV calling and phasing[47]. VolcanoSV then partitions reads into the corresponding haplotype within each phase block. Leveraging a Pair-Hidden Markov Model (pair-HMM) and the HapCUT2 algorithm[48] for read-based haplotype estimation, Longshot has demonstrated exceptional utility in mitigating the challenges posed by the high error rates inherent to long sequencing reads[47]. In the total 14 long reads datasets we investigated, approximately 75% of the reads are assigned to a certain phase block and haplotype, while a substantial proportion of the reads are still not assigned to any phase block (Fig. 1). We referred to these intractable reads from Longshot as "unphased reads". These reads often do not cover enough or any heterozygous SNVs for a haplotype phasing algorithm to resolve them. We thus designed a new algorithm to partition them into two haplotypes within a certain phase block.

**Unphased reads assignment through unique *k*-mer similarity analysis.** To accurately assign the unphased reads to the corresponding haplotype, a unique *k*-mer similarity-based cost-efficient approach is designed (Fig. 1). The underlying mechanism relies on the fact that each haplotype (within each phase block) is composed of distinct sets of mutation events (SNPs, small indels, and SVs) and these events allow us to extract a haplotype-specific unique *k*-mer set to represent it. VolcanoSV can then assign unphased reads to the specific haplotype by comparing the correspondence between each unphased read and nearby haplotypes relying on unique *k*-mer sets similarity. iVolcanoSV utilizes every two adjacent phase blocks of phased reads to define unique *k*-mers and extract haplotype-specific unique *k*-mer sets for all four haplotypes. Unique *k*-mers are defined as ones only appearing in one of four haplotypes. For each unphased read, VolcanoSV also extracts *k*-mers and then quantifies the percentage of its unique *k*-mers which are assigned to each haplotype of two adjacent phase blocks. If an unphased read is originally drawn from one specific haplotype, it is expected to see a high correspondence between the unphased read and the haplotype. VolcanoSV uses an empirical distribution quantile-based significance test to quantify the correspondence based on four calculated percentages and then assigns each unphased read. If an unphased read cannot be assigned when the significance test does not pass the criterion, VolcanoSV assumes this read to be drawn from both haplotypes and assigns it to both haplotypes of its nearest phase block. At the end of this module, unphased reads are partitioned to the corresponding haplotype for each phase block. The detailed methods are described as follows:

Firstly, VolcanoSV assigns each unphased read to its candidate phase blocks. The criteria for determining the candidate phase block are as follows:

- If the unphased read overlaps with the phase block(s) according to the coordinates, the overlapping phase block(s) will be assigned as its candidate phase block(s).
- If the unphased read does not overlap with any phase block, but is in the gap between two consecutive phase blocks, these two phase blocks will be assigned as candidates.
- If the above two conditions do not apply, i.e., the unphased read is aligned to the start or end segment of a chromosome and does

not overlap with any phase block, the nearest phase block will be assigned as the candidate.

As a result, every unphased read is assigned to at least one phase block and two PS_HPs. "PS_HP" refers to one haplotype of a phase block in the following context.

Secondly, to determine which PS_HP (haplotype of a phase block) the unphased read is drawn from, a unique $k$-mer similarity-based analysis is performed. VolcanoSV first collects a raw $k$-mer set for every candidate PS_HP. The raw $k$-mer set for a PS_HP is the union of $k$-mers from all phased reads belonging to this PS_HP. When collecting $k$-mers, the length of kmer is set to 12, and step size is set to 1 by default. Next, VolcanoSV creates "fingerprint" (unique) $k$-mer sets that are exclusive to every PS_HP. The fingerprint $k$-mer set of a PS_HP is defined as the intersection between the raw $k$-mer set of this PS_HP and the symmetric difference among all candidates PS_HPs. For example, if an unphased read has 4 candidate PS_HPs, of which the raw $k$-mer sets are R1, R2, R3, R4, then the symmetric difference among them is

$$
\begin{aligned}
\Delta(R1, R2, R3, R4) = &(R1 \cup R2 \cup R3 \cup R4) \\
&-(R1 \cap R2) - (R1 \cap R3) \\
&-(R1 \cap R4) - (R2 \cap R3) \\
&-(R2 \cap R4) - (R3 \cap R4)
\end{aligned} \tag{1}
$$

The fingerprint (unique) $k$-mer set of each PS_HP is defined as follows

$$
\begin{cases}
F1 = R1 \cap \Delta(R1, R2, R3, R4) \\
F2 = R2 \cap \Delta(R1, R2, R3, R4) \\
F3 = R3 \cap \Delta(R1, R2, R3, R4) \\
F4 = R4 \cap \Delta(R1, R2, R3, R4)
\end{cases} \tag{2}
$$

Denoting the $k$-mer set of the unphased read as S, the unique $k$-mer similarity metrics between the unphased read and the four candidates PS_HPs are then defined as the size of their $k$-mer sets intersections.

$$
\begin{cases}
Sim1 = |S \cap F1| \\
Sim2 = |S \cap F2| \\
Sim3 = |S \cap F3| \\
Sim4 = |S \cap F4|
\end{cases} \tag{3}
$$

The normalized similarity metrics for the four candidates PS_HPs are calculated as follows

$$
NormSim_i = \frac{Sim_i}{\sum_{i=1}^{4} Sim_i} \tag{4}
$$

VolcanoSV repeats this procedure for all unphased reads and their candidate PS_HPs. VolcanoSV thus collects all normalized similarity metrics, forming a normalized similarity vector $\chi$. To finally determine which PS_HP the unphased read is drawn from, VolcanoSV utilizes an empirical distribution quantile-based significance test to evaluate the normalized similarity metrics between unphased reads and candidate PS_HPs. A level $r$ (10% by default) is used, and the cut-off threshold for significance is the $(1 − r)$ quantile of the normalized similarity vector $\chi$. Metrics exceeding this threshold are considered significant, and reads are assigned accordingly. The null hypothesis (H0) posits that the normalized similarity metric between the unphased read and the candidate PS_HP is not significantly different from what would be expected by random chance, i.e., $NormSim_i \le Q_{1-r}(\chi)$. For each unphased read, we compare its normalized similarity metric to the cut-off $Q_{1-r}(\chi)$. If the metric is higher than the cut-off, it is considered significant, suggesting a potential association with the corresponding PS_HP. Conversely, if a normalized similarity metric for an unphased

read does not exceed the cut-off, we fail to reject the null hypothesis for that specific read and candidate PS_HP combination, implying that there is no significant association and the observed similarity might be due to random chance. If an unphased read can not be assigned to any candidate PS_HP based on the significance test, it will be partitioned to both haplotypes of its nearest phase block.

**Haplotype-aware local assembly via augmented phase blocks.** Once all unphased reads are partitioned to the corresponding haplotype of a certain phase block, VolcanoSV performs haplotype-aware local assembly on all partitioned reads for each haplotype of all augmented phase blocks. For PacBio Hifi data, VolcanoSV employs hifiasm (v0.14)[26] to perform the local assembly. For PacBio CLR or ONT data, Flye[27] is utilized for assembly in this study. Additionally, users have the flexibility within this module to choose the most optimal or new assemblers to fulfill their specific requirements. We also added functionality allowing users to select specific assemblers for target regions defined by a BED file. At the end of this module, haplotype-resolved contiguous sequences (contigs) are generated for each phase block.

**Contig alignment-based signature collection for large indel SV detection.** To detect SVs based on haplotype-resolved contigs, VolcanoSV performs a contig-to-reference alignment using minimap2 (v2.24-r1122)[39]. VolcanoSV then collects large insertion (INS), deletion (DEL), inversion (INV), translocation (TRA), and duplication (DUP) signatures relying on the contigs alignment BAM file. For small insertions and deletions (indels), signatures are usually inferred through the CIGAR field of the BAM file (intra-alignment), while the signatures of large INS, DEL, INV, TRA, and DUP can only be inferred through split alignment (inter alignment). Specifically, for large indel SVs, VolcanoSV adapts and optimizes the reads signature methods from traditional reads alignment-based tools[17,49] to haplotype-resolved contigs alignment to collect signatures.

Large INSs are detected when two disjoint segments on a contig are aligned to two adjacent segments on the reference genome in the same orientation. Likewise, large deletions are inferred when two adjacent segments on a contig are aligned to two disjoint segments on the reference genome with a gap in between. We denote the aligned segment's start and end coordinates relative to a contig as Contig_s(start) and Contig_e(end), and relative to the reference as Ref_s(start) and Ref_e(end). These notations are also illustrated in the pipeline Fig. 2. Two metrics can be defined using the start and end coordinates of a pair of split-alignments from a single contig (two segments from the split-alignment are referred to as indices 1 and 2 in the metrics below).

$$
\begin{cases}
Diff\_dis = (Contig\_2s − Contig\_1e) \\
\qquad\qquad − (Ref\_2s − Ref\_1e) \\
Diff\_olp = Ref\_1e − Ref\_2s \text{ for INS} \\
\qquad\qquad \text{or} \\
Contig\_1e − Contig\_2s \text{ for DEL}
\end{cases} \tag{5}
$$

Diff_dis is defined as the difference between the distance of two segments on the contig and their distance on the reference. Specifically, in the presence of an INS, the distance between the two segments on the contig will be greater than their distance on the reference because a subset of the contig sequence can not be continuously aligned to the reference. In the case of a DEL, the situation is reversed due to the loss of sequence from the contig compared to the reference. In practice, Diff_dis should be approximately equal to the size of SVs. Therefore, we defined an empirical threshold parameter for Diff_dis, denoted as $TH_{olp}$, which is set to 30 bp by default. This ensures the signature VolcanoSV generates will adequately cover any SV larger than or equal to 50 bp.

Diff_olp represents the overlapping portion of the alignment between two segments. For INSs, this alignment occurs on the reference sequence, whereas for DELs, it occurs on the contig. The value of Diff_olp should not be excessively large, even in the presence of a large INS or DEL. In the case of INSs, if two non-contiguous segments flanking the inserted sequence are accurately aligned to the reference, their alignment records should be contiguous on the reference. A significant overlap on the reference, despite a large Diff_dis, still suggests a false positive. Similarly, for deletions, if the two segments on the contig exhibit a substantial overlap, this likely indicates a false positive. To mitigate false positives, a threshold parameter for Diff_olp, denoted as TH_olp, has been introduced and is set to 3000 bp by default.

So, VolcanoSV decides a large INS exits if the following conditions apply

$$
\begin{cases}
\text{Diff\_dis} \geq \text{TH}_{dis} \\
\text{Diff\_olp} < \text{TH}_{olp}
\end{cases}
\tag{6}
$$

or a large DEL exits if the following conditions apply

$$
\begin{cases}
\text{Diff\_dis} \leq -\text{TH}_{dis} \\
\text{Diff\_olp} < \text{TH}_{olp}
\end{cases}
\tag{7}
$$

If the rule applies, the INS signature is collected by VolcanoSV as follows

$$
\begin{cases}
\text{chromosome} = \text{contig\_aligned\_reference\_name} \\
\text{start} = (\text{Ref\_1e} + \text{Ref\_2s})/2 \\
\text{svlen} = |\text{Contig\_2s} - \text{Contig\_1e} + \text{Diff\_olp}|
\end{cases}
\tag{8}
$$

The DEL signature is collected by VolcanoSV as follows

$$
\begin{cases}
\text{chromosome} = \text{contig\_aligned\_reference\_name} \\
\text{start} = \text{Ref\_1e} \\
\text{end} = \text{Ref\_1e} + |\text{Diff\_dis}| \\
\text{svlen} = \text{Diff\_dis}
\end{cases}
\tag{9}
$$

**Clustering and refinement of large indel SV signatures on each haplotype.** Large indel SV signatures are collected by VolcanoSV on contigs from each haplotype separately. Ideally, the contig depth for one haplotype should be equal to 1 for any position and every signature should be disjoint perfectly. However, in practice, due to assembly artifacts and contig alignment issues, there may be overlapping contigs aggregated at certain positions, which could cause the signatures to have redundancy. Therefore, VolcanoSV employs a clustering algorithm to refine signatures collected from each haplotype. The clustering procedure is illustrated in the pipeline Fig. 2. In the first step, signatures are divided into two categories, INS and DEL, and are sorted by breakpoint positions respectively. Next, VolcanoSV uses a nearest-neighbor chain algorithm to cluster (merge) the signatures of each category. To achieve this, each signature is considered as a node, if the similarity between two adjacent nodes passes a certain similarity threshold, an edge is added between them. VolcanoSV calculates the similarity metrics from the first to the last pair of signature nodes in the sorted list. As a result, a substantial number of disjoint subclusters are generated and each subcluster represents an SV call. In most cases, the subcluster contains only one signature, which is directly selected as the actual SV call. In the few cases where a subcluster contains more than one signature, VolcanoSV selects the signature with the largest length as the actual SV call. In more detail, the similarity metric between each pair of INS signatures is computed as follows (Each pair

of signatures is referred to as indices 1 and 2 in the metrics below):

$$
\begin{cases}
\text{breakpoint\_shift} = \text{start}_1 - \text{start}_2 \\
\text{svlen\_sim} = \frac{\min(\text{svlen}_1,\text{svlen}_2)}{\max(\text{svlen}_1,\text{svlen}_2)}
\end{cases}
\tag{10}
$$

The condition for two INS signature nodes to be clustered together is

$$
\begin{cases}
\text{breakpoint\_shift} < \text{TH}_{shift\_intrahap} \\
\text{svlen\_sim} > \text{TH}_{sim}
\end{cases}
\tag{11}
$$

where $\text{TH}_{shift\_intrahap}$ is a distance threshold parameter used to merge any two INS signatures from the same haplotype. VolcanoSV sets this threshold to 100 bp by default to ensure a stringent merging of signatures within the same haplotype. $\text{TH}_{sim}$ is another threshold for SV similarity, specifically SV length, used to merge any two INS signatures from the same haplotype. Once a stringent $\text{TH}_{shift\_intrahap}$ is applied, VolcanoSV sets a moderately tolerant threshold for $\text{TH}_{sim}$, which is 0.5 by default.

The similarity between each pair of DEL signatures is measured as follows:

$$
\begin{cases}
\text{breakpoint\_shift} = \text{start}_1 - \text{start}_2 \\
\text{svlen\_sim} = \frac{\min(\text{svlen}_1,\text{svlen}_2)}{\max(\text{svlen}_1,\text{svlen}_2)} \\
\text{overlap\_ratio} = \frac{(\min(\text{end}_1,\text{end}_2) - \max(\text{start}_1,\text{start}_2))}{\min(\text{svlen}_1,\text{svlen}_2)}
\end{cases}
\tag{12}
$$

The condition for two DEL signature nodes to be clustered together is

$$
\begin{cases}
\text{breakpoint\_shift} < \text{TH}_{shift\_intrahap} \\
\text{svlen\_sim} > \text{TH}_{sim} \\
\text{overlap\_ratio} > \text{TH}_{sim}
\end{cases}
\tag{13}
$$

The default threshold values of $\text{TH}_{shift\_intrahap}$ and $\text{TH}_{sim}$ for DEL are the same as those for INS. For any two DEL signatures, VolcanoSV also evaluates their overlapping ratio and uses the same similarity threshold parameter, $\text{TH}_{sim}$.

**Pairing large indel SV calls from both haplotypes and genotype prediction.** After collecting and clustering signatures from each haplotype, a pairing algorithm is applied to merge large indel SV calls and determine the genotype. SV calls from each haplotype are first merged and sorted by position, for INS and DEL SVs respectively. Next, a chaining procedure similar to the previous within-haplotype signature clustering is performed on the sorted SV list under a different threshold setting. Specifically, an edge will be added between two INS SVs if the following condition applies:

$$
\begin{cases}
\text{breakpoint\_shift} < \text{TH}_{shift\_interhap} \\
\text{svlen\_sim} > \text{TH}_{sim}
\end{cases}
\tag{14}
$$

An edge will be added between two DEL SVs if the following condition applies:

$$
\begin{cases}
\text{breakpoint\_shift} < \text{TH}_{shift\_interhap} \\
\text{svlen\_sim} > \text{TH}_{sim} \\
\text{overlap\_ratio} > \text{TH}_{sim}
\end{cases}
\tag{15}
$$

Compared to intra-haplotype clustering, the distance threshold for inter-haplotype clustering is relatively more relaxed. $\text{TH}_{shift\_interhap}$ is set to 200 bp by default because more influencing factors, such as assembly artifact and alignment ambiguity, are expected to affect the signatures between two haplotypes. The SV similarity threshold, specifically for SV length and DEL overlapping ratio, remains the same as in intra-haplotype clustering, setting to 0.5 by default.

As a result, a substantial number of subclusters is generated to represent INS and DEL SVs, respectively. Similar to the procedure in within-haplotype signature clustering, the largest SV call in each subcluster is kept. In addition, if a subcluster contains SVs from both haplotypes, the genotype is labeled as "1|1". If a subcluster contains SVs from only one haplotype, the genotype is labeled as "0|1" or "1|0" depending on which haplotype the SVs are drawn from with a specific phase block.

**Large indel SV filtering.** Inferred from the contig-to-reference alignment file, the raw SV calls might contain some noise or artifacts introduced by the assembly and alignment procedure. Therefore, a filtering algorithm based on read alignment information is designed to remove false calls from raw SV calls.

For large indel SVs, similar to the procedure of collecting signatures from the contig-to-reference alignment file, reads-based SV signatures are first collected by scanning the read-to-reference file. Next, for each raw SV call, VolcanoSV scans through all read-based signatures of the same type (either INS or DEL) within the 1 kb region centered around the inferred breakpoint of the raw SV and calculates the SV length similarity. Denoting the size of raw SV as $svlen_{raw}$ and the size of read-based signature as $svlen_{read}$, the SV length similarity between them is calculated as follows

$$svlen\_sim = \frac{min(svlen_{raw}, svlen_{read})}{max(svlen_{raw}, svlen_{read})} \qquad (16)$$

If the SV length similarity is greater than the minimum SV length similarity threshold (0.5 by default), this read-based signature will be regarded as supporting the raw SV. A raw SV call needs at least one supporting signature, otherwise it will be filtered out as a false call. Since the read-based alignment information is more reliable for small-size SVs, we only apply this filtering procedure for indel SV signatures within the size range of 50–250 bp.

Compared to INS calls, whole-genome assembly-based methods like VolcanoSV often generate more false positive DEL calls. To further remove false positives in DEL calls, a more stringent filtering method is required. VolcanoSV first scans through read-based signatures and collects all DEL signatures that are within the 1 kb flanking region around the breakpoint of each candidate DEL call. Next, VolcanoSV calculates the supporting signature depth for each candidate DEL call in the following fashion:

$$sig\_depth = \frac{\sum_{i=1}^{N} svlen\_sig_i}{svlen_{candidate}} \qquad (17)$$

where $svlen\_sig_i$ is SV length of the $i$th supporting signature within the flanking region, and $N$ is the number of supporting signatures in the flanking region. By collecting the supporting signature depth metrics for all candidate DELs, VolcanoSV obtains a vector **R**. A DEL SV is considered true if the below condition applies:

$$lb \cdot \tilde{\mathbf{R}} \leq sig\_depth \leq rb \cdot \tilde{\mathbf{R}} \qquad (18)$$

where $\tilde{\mathbf{R}}$ is the median value of **R**, and $lb$ and $rb$ are the left and right boundary ratios, which are dependent on the long reads data type. For example, for Hifi data, $lb = 0.2$ and $rb = 2.6$; For CLR data, $lb = 0.19$ and $rb = 3.0$; for ONT data, $lb = 0.24$ and $rb = 2.8$. Those parameters are drawn empirically.

**Remove redundancy.** To further remove the redundancy in the filtered SV call set, we designed a rigorous one-to-$K$ clustering algorithm to identify duplicate SV calls in which SV calls are first separated by category and sorted by position. VolcanoSV then compares each SV and its $K$ neighboring SVs (within the flanking region of 500 bp) and

calculates similarity metrics. VolcanoSV adds an edge between this SV and any neighboring SV that passes the similarity threshold. The similarity metric includes breakpoint distance, SV length similarity, and SV sequence similarity (edit distance).

For INS SVs, the threshold for adding an edge between two calls is

$$\begin{cases} breakpoint\_shift < TH_{shift\_redun\_INS} \\ svlen\_sim > TH_{sim} \\ seq\_sim > TH_{sim} \end{cases} \qquad (19)$$

For DEL SVs, the threshold for adding an edge between two calls is

$$\begin{cases} breakpoint\_shift < TH_{shift\_redun\_DEL} \\ svlen\_sim > TH_{sim\_relax} \\ seq\_sim > TH_{sim} \end{cases} \qquad (20)$$

where $TH_{shift\_redun\_INS}$ and $TH_{shift\_redun\_DEL}$ are set to 500 bp and 300 bp, respectively. The SV similarity threshold parameter, $TH_{sim}$, retains the default setting of 0.5. For the DEL length similarity threshold, $TH_{sim\_relax}$ is set to 0.1 by default. The distance threshold is more relaxed compared to previous intra- and inter-haplotype clustering to further remove redundancy that was overlooked in the earlier clustering steps. $TH_{shift\_redun\_DEL}$ is more stringent than $TH_{shift\_redun\_INS}$, while the svlen_sim threshold for DEL is more relaxed than for INS, reflecting observed redundancy distribution patterns: redundant INSs tend to be further apart with highly consistent sizes, whereas redundant DELs are typically closer together with more variable sizes. These empirically derived thresholds ensure that VolcanoSV achieves both sensitive and highly accurate clustering.

After all edges are added, VolcanoSV selects the SV call of the largest SV length as the final prediction for each subcluster. This one-to-$K$ clustering method is more extensive than previous intra and inter-haplotype clustering because it is more tolerant to noise or false calls when chaining consecutive SVs. For example, let's consider three consecutive SV calls: a 1 kb INS, 60 bp INS (very likely a false call), and 1.01 kb INS. In the previous chaining algorithm, since the second call is not similar to either the one before or the one after, the chain would be broken and the two otherwise very similar INS calls can not be clustered together. However, they will be clustered together in the one-to-K cluster algorithm since any SV pairs within a certain size range will be compared. In addition, this clustering method is also rigorous by taking sequence similarity into consideration to avoid false clustering.

**Genotype refinement.** Since every contig produced by VolcanoSV's pipeline is associated with a certain phase block and a haplotype, it is straightforward to pair SV calls from contigs of both haplotypes within a phase block to determine the phased genotype of SV calls. VolcanoSV applies a heuristic decision tree model to further refine the genotype for large indel SVs.

Specifically, VolcanoSV takes five parameters as input: genotype inferred by contigs (0|1 or 1|1), SV size (binary variable, equal to 1 if the SV size is greater than 1 kb, otherwise equal to 0), SV type (DEL or INS), long reads sequencing technology (Hifi, CLR or ONT), and the relative supporting read-based signature ratio (number of the supporting read-based signatures/local read depth). VolcanoSV then uses an empirical threshold to predict the genotype. In more detail, the first four parameters are categorical and they define a tree structure with 24 leaf nodes, while the last parameter, the relative supporting signature ratio, is a continuous variable ranging from 0 to 1. Ideally, the relative supporting signature ratio should be tightly associated with the actual genotype, i.e., the relative ratio should be close to 0.5 for heterozygous SVs, and 1 for homozygous SVs. However, in practice, this relative ratio varies due to complex confounding factors (SV type, SV size, sequencing technology, etc). Therefore, VolcanoSV applies 24 different thresholds with respect to the relative supporting signature ratio, with

the 24 leaf nodes. For any large indel SV, given the first four parameters, VolcanoSV will first decide the leaf node and the corresponding threshold for the relative supporting ratio, then use the threshold and the observed relative supporting ratio to predict the genotype.

**Complex SV collection, recovery, and filtering.** With respect to complex SVs including inversions (INVs), translocation (TRAs), and duplications (DUPs), VolcanoSV integrates the signature collection pipeline from SVIM-asm[24] and implements a specific duplication recovery and complex SV filtering pipeline to generate the final complex SV calls.

INVs are inferred when two adjacent segments on the contig are aligned to the reference genome in different orientations. To filter false calls, VolcanoSV first adds a 1 kb flanking region around each INV, and then investigates aligned reads information around both the start and end position of the INV. In either breakpoint position, if there exists at least one read that is aligned to the reference genome in two distinct orientations, this inversion signature will be then kept, otherwise, it will be removed as a false call.

TRAs are inferred when two segments from a contig are aligned to two different chromosomes. To filter false calls, VolcanoSV first adds a 1 kb flanking region around each TRA. VolcanoSV then investigates the aligned reads information around both breakends (BND) of the TRA. If a certain amount of reads ($\geq 0.25*$read depth) is aligned to both breakends, the TRA call will be kept, otherwise it will be filtered out as a false call.

DUPs are inferred when two adjacent segments on the contig are aligned to identical or overlapping segments on the reference genome. For DUPs, the scenario is very different from the other SV types. Due to the limitation of the current alignment method, a significant amount of DUPs in the contigs are in fact treated as INSs in the alignment practice. As a result, DUP events are most often represented either by an insertion cigar or a split-alignment that leads to INS prediction. Adapted to this scenario, VolcanoSV utilizes a specific pipeline to recover the missed DUP calls from INS calls. Specifically, VolcanoSV extracts the alternate allele sequences of all INS calls and aligns them back to the reference genome. If an inserted sequence is aligned back to a position that is close to the insertion breakpoint, then this insertion call is recovered as a DUP call. Through this recovery procedure, VolcanoSV achieves a comprehensive and reliable duplication discovery.

**Small indel collection and refinement.** For small (2–49 bp) indel calling, VolcanoSV integrates a signature collection pipeline from Dipcall. This pipeline scans through the contig-to-reference BAM file to gather the indel signatures. VolcanoSV then utilizes a customized $k$-mer content analysis to filter and refine the indel calls. In more detail, after collecting the indel signatures from the contig-to-reference BAM file, VolcanoSV first extracts the indel context sequence (with an additional 20 bp flanking region around each indel) directly from the contig. Subsequently, for each indel, VolcanoSV assesses the $k$-mer content from the reads-aligned BAM file within the 100 bp region centered around the indel breakpoints. For a candidate indel, if more than 70% of the kmers in its contig context are also present in the $k$-mer content of the reads from the corresponding region, this indel will be counted as a high confidence call; otherwise, it will be regarded as an assembly or alignment artifact and will be filtered out. For SNP calling, VolcanoSV by default provides the phased SNP call results inferred by Longshot.

### Benchmarking SV calls using Truvari

The GIAB community provides a gold standard SV set for the HG002 sample[50], which includes 4117 deletions and 5281 insertions in defined "high-confidence" regions characterized by multiple

sequencing platforms. SV calls (deletions and insertions) from all tools were evaluated against this benchmark using Truvari[30], which is a commonly used open-source toolkit for the comparison, annotation, and analysis of structural variation.

Truvari provides metrics/parameters including pctstim ($p$), pctsize ($P$), pctovl ($O$), and refdist ($r$) to set different criteria for SV evaluation depending on the needs of the specific analysis. The parameter $p$ controls the minimum allele sequence similarity used to identify two SV calls as identical. The similarity is calculated from the edit distance ratio between the reference and alternate haplotype sequences of the base and comparison call. Setting $p$ to zero can disable this comparison. The parameter $P$ corresponds to the minimum allele SV length similarity between the compared SVs, which is calculated by dividing the length of the shorter SV with the longer one. The parameter $O$ determines the minimum threshold of the reciprocal overlap ratio between the base and comparison call, and it is only applied to deletions for evaluating the effect of breakpoint shift on deletion accuracy. The parameter $r$ represents the threshold for maximum reference location difference of the compared SVs, which can be used to evaluate the effect of breakpoint shift on insertion accuracy. In general, higher values of $p$, $P$ and $O$, and lower values of $r$ set more stringent comparison criteria, as they will require the compared SVs to have higher sequence and SV length similarity, larger spatial overlapping ratio, or closer location to the reference sequence to be considered as the same SV.

### SV breakpoint shift and alternate allele sequence similarity analysis

Breakpoint shift is calculated from the reference genome location difference between the true positive SVs called by the tools and the corresponding benchmark SVs. Called SVs and benchmark SVs are paired up by the "MatchID" provided by Truvari. For each deletion, the start and end coordinate differences between the called SV and benchmark SV are calculated, and the maximum value of these two is chosen as the value for breakpoint shift. For insertions, the breakpoint shift is defined as the start coordinate difference. Breakpoint shift values larger than 200 bp are merged to the 200+bp bin in the distribution plot.

### Annotating and visualizing false discoveries on the T2T-CHM13 reference

We used the UCSC Genome Browser[51,52] to load various tracks for annotating and visualizing false discoveries identified by four assembly-based tools. To annotate heterochromatin regions[53,54], such as centromeres and telomeres, in the T2T-CHM13 reference, we downloaded two BED files from https://s3-us-west-2.amazonaws.com/human-pangenomics/T2T/CHM13/assemblies/annotation/chm13v2.0_censat_v2.0.bed and https://s3-us-west-2.amazonaws.com/human-pangenomics/T2T/CHM13/assemblies/annotation/chm13v2.0_telomere.bed. These BED files were merged and intersected with the false discoveries using bedtools[55].

```
bedtools intersect -a ${FD_VCF} -b ${Telo_CenSat_BED}
-u > ${out_VCF}
```

### Assessing diploid assembly using Flagger and annotating regions enriched in segmental duplications

To assess the quality of our assemblies, we employed a read-based pipeline, Flagger[45], to annotate the contig regions. Flagger identifies different types of misassemblies within a phased diploid assembly. The pipeline performs haplotype-aware mapping of long reads to the combined maternal and paternal assembly. It identifies potential assembly errors by pinpointing coverage inconsistencies in these mappings.

In this study, we utilized the basic version of Flagger (flagger_end_to_end_no_variant_calling_no_ref_no_secphase.wdl, which can be

found at https://github.com/mobinasri/flagger/releases/tag/v0.3.3). The input files included the read-to-contig BAM file and the contig FASTA file. The primary input parameters were maxReadDivergence and window size. For maxReadDivergence, we set the value to 0.02 for HiFi and 0.09 for ONT and CLR data, as recommended by the Flagger GitHub manual (https://github.com/mobinasri/flagger). The default window size used by Flagger is 5 MB, which is suitable for most whole-genome scale assemblies. However, since the VolcanoSV-asm pipeline is based on a local assembly strategy, we adjusted the window size to 150 kb to accommodate all contigs, including those shorter than 5 MB. With these input files and parameter settings, Flagger fits the Gaussian mixture model and generates a BED file with contig regions annotated by four labels: haploid, error, (falsely) duplicated, and collapsed. Regions labeled as haploid are expected to have error-free assemblies. The erroneous component, modeled by a Poisson distribution, represents the regions with very low read support. The (falsely) duplicated component is characterized by regions with only half of the haploid component's mean coverage. The collapsed component's mean coverage is constrained to be multiples of the haploid component's mean. We utilized collapsed components to annotate potential misassemblies enriched in segmental duplications (SDs).

To assess the SD reliability of the Flagger annotation, we aligned the regions of collapsed components from diploid assemblies by VolcanoSV to the GRCh38 reference genome and intersected them with the SD annotations for HG002 based on GRCh38. The HG002 SD annotation file was downloaded from https://ftp-trace.ncbi.nlm.nih.gov/ReferenceSamples/giab/release/genome-stratifications/v3.4/. We calculated and compared the total length of SD annotation regions that overlap with collapsed regions by Flagger across all VolcanoSV assemblies, which were generated using different assemblers including wtdbg2, Shasta, NextDenovo, and Hicanu.

### Reporting summary

Further information on research design is available in the Nature Portfolio Reporting Summary linked to this article.

## Data availability

PacBio CLR, Hifi, and ONT sequencing reads for HG002 are available at GIAB and NCBI. The high-confidence HCC1395 somatic SV callset and the Pacbio and ONT Tumor-Normal paired libraries of HCC1395 are publicly accessible at NCBI. PacBio Hifi sequencing reads for CHM13 are available at NCBI. Table 1 lists hyperlinks for all 20 previously mentioned real datasets. The Tier1 benchmark SV callset and high-confidence HG002 region were obtained from https://ftp-trace.ncbi.nlm.nih.gov/ReferenceSamples/giab/data/AshkenazimTrio/analysis/NIST_SVs_Integration_v0.6/. T2T assembly is publicly available at https://github.com/marbl/CHM13. The HG002 SD annotation file was downloaded from https://ftp-trace.ncbi.nlm.nih.gov/ReferenceSamples/giab/release/genome-stratifications/v3.4/. All assembled contig files and VCF files that support the findings of this study are available from https://doi.org/10.5281/zenodo.10456757. Data and code required to reproduce the results presented in this study are available in the Source Data file. Source data are provided with this paper.

## Code availability

All code is available at https://github.com/maiziezhoulab/VolcanoSV under the MIT License[56].

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

## Acknowledgements

This work was supported by the NIH NIGMS Maximizing Investigators' Research Award (MIRA) R35 GM146960 to X.M.Z.

## Author contributions

X.M.Z. conceived and led this work. C.L and X.M.Z. designed the framework. C.L. implemented the framework, and C.L. and Y.H.L. performed data analysis. C.L, Y.H.L, and X.M.Z. wrote the manuscript.

## Competing interests

The authors declare no competing interests.
