## [Peer Review File · Nature Communications]

VolcanoSV enables accurate and robust structural variant calling in diploid genomes from single-molecule long read sequencingREVIEWER COMMENTS

Reviewer #1 (Remarks to the Author):

Summary: In this manuscript, Luo et al. present VolcanoSV, which is a software package implementing a novel assembly algorithm for SV calling using long-read DNA sequence data. Their description of the algorithm is suitably detailed and the comparison tests they perform relative to other assembly-based SV tools are promising. However, I feel that their comparisons suffer greatly from a cherry-picked context relative to alignment-based SV calling methods. Namely, they choose to highlight ambiguity in breakpoint detection as the major weakness of alignment-based methods without exploring the precision and recall advantages such methods may have against their own. I recommend that comparative tests must be complete and thorough or that this comparison should be fully dropped from the text in subsequent revisions.

Here are my comments in no particular order:

Page 9: This is the first use of the acronym (PS_HP) and it should be better defined in the text of future context. I am assuming that it stands for "Phased Haplotype."

Page 10: It would be useful to mention why there is a difference in assembly algorithm used for HiFi and other long read datasets. Flye has an input method for HiFi reads, for example.

Figure 3: The caption does not mention that the UpSet plot shows the overlap of true positive SVs detected from the assembly method test, but it should to improve the readability of this section.

Page 5: The recall ratios for the high-confidence tumor SV calls is quite low. Is there any association of the likelihood of detection with the methods of original discovery used in the original manuscript? For example, is VolcanoSV more likely to detect translocations originally identified by other long read sequence datasets?

Page 5: I feel that the comparison of assembly- and alignment-based SV callers is incomplete. The authors appear to try to highlight the strengths of assembly-based methods by only using their alignment-based callsets in a parameter-optimization exercise to demonstrate the inability of these tools to generate precise breakpoints. However informative this may be, it is not a fair demonstration. I would prefer to see F1 and Recall statistics from the alignment-based methods on the same GIAB, simulation, and tumor datasets to appropriately compare the results of this test.

Page 7: The T2T assemblies often have assembled heterochromatin regions that are difficult to assemble apart from the use of specialized graph-based methods. Are the false positive calls from the assembly-based tools localized to some of these heterochromatin regions? Is that where the main source of error exists?

Page 7: One of the distinct advantages of alignment-based variant calling methods is that they often have lower computational footprints. A discursive analysis is not required here, but what is the relative difference in computational time and resources of Volcano-SV vs alignment based methods of SV discovery?

Github repository: The authors include an unlinked version of HifiASM in their repository. Proper credit to the original authors of the repo should be given, or a hard link to that specific version of the repo should be added to the main repository for VolcanoSV

Reviewer #1 (Remarks on code availability):

The code base has a suitable README and is documented with some quick start instructions. It seems suitable to test and run the software package as presented. I note some discrepancies in code reuse from other software packages in my comments to the Authors.

Reviewer #2 (Remarks to the Author):

The manuscript introduces VolcanoSV, a novel hybrid SV detection pipeline that enhances the precision of identifying structural variants (SVs) in the human genome. The tool combines reference genome and local de novo assembly methods to accurately pinpoint SV breakpoints and sequences. Despite its promising potential, the manuscript requires significant revision to address certain issues and improve clarity and completeness in presenting VolcanoSV's methodology and findings. Below are the major points that need to be addressed:

VolcanoSV incorporates existing tools for sequence assembly, which may not always be the most suitable choice. Particularly in regions rich in segmental duplications, it is recommended to employ tools that outperform Flye in handling such complexities.

The validation scope of the method is somewhat limited, and the metrics obtained for some competitors are very similar, with differences only evident after the second decimal place. The manuscript would benefit from demonstrating the implications of such improvements, for example, showing how a 0.59% increase in GT accuracy translates into tangible benefits or impacts on downstream analyses. The Truvari toolkit should be more thoroughly described, with its parameters clearly explained to enhance understanding.

The methodology is not clearly described. The notation is rather unclear, and inconsistent font usage further complicates readability. What is the null hypothesis in the chi-square test used in unphased reads assignment? Is the similarity defined in equation (4) a metric? On what basis were the thresholds selected in equation (7) (13) (14) (19) and (20)?

The sentence: "In an ideal case, when the subcluster only contains one signature, this signature is directly selected as the actual SV call" should be accompanied by the statistics of how this looks in general. In equation (12), it is unclear why the absolute value pertains to only one sequence, and additionally, the parentheses are mismatched.

The figures are very unclear. Fig. 1 should be divided into several parts, while Figure 2 does not add any value and could be presented as a table instead.

Reviewer #2 (Remarks on code availability):

This GitHub resource offers clear documentation, commented code, and practical examples, making it accessible for developers.

Editor

We appreciated the expert review and overall positive feedback of our article titled “VolcanoSV enables accurate and robust structural variant calling in diploid genomes from single-molecule long read sequencing”. We have made extensive changes to the manuscript to address all issues raised. We believe the revised version is substantially strengthened as a result.

Reviewer #1 (Remarks to the Author):

Summary: In this manuscript, Luo et al. present VolcanoSV, which is a software package implementing a novel assembly algorithm for SV calling using long-read DNA sequence data. Their description of the algorithm is suitably detailed and the comparison tests they perform relative to other assembly-based SV tools are promising. However, I feel that their comparisons suffer greatly from a cherry-picked context relative to alignment-based SV calling methods. Namely, they choose to highlight ambiguity in breakpoint detection as the major weakness of alignment-based methods without exploring the precision and recall advantages such methods may have against their own. I recommend that comparative tests must be complete and thorough or that this comparison should be fully dropped from the text in subsequent revisions.

We thank Reviewer 1 for the positive evaluation towards the potential of this new method and for the insightful comments on this study. We have performed additional analyses to further improve the rigor of this method. All the changes are highlighted in red in the main manuscript and supplementary information.

Here are my comments in no particular order:

- 1. Page 9: This is the first use of the acronym (PS_HP) and it should be better defined in the text of future context. I am assuming that it stands for “Phased Haplotype.”*

Response: The reviewer’s point is well taken. We clarified and defined “PS_HP” in the corresponding text of the Methods section.

2. *Page 10: It would be useful to mention why there is a difference in assembly algorithm used for HiFi and other long read datasets. Flye has an input method for HiFi reads, for example.*

Response: The reviewer's point is well taken. Since this point aligns partially with major point 1 raised by Reviewer 2, we have included a justification for the default assemblers utilized in VolcanoSV. This is detailed in a newly added Results section titled "Selecting assemblers for regions rich in segmental duplications".

We have provided the justification below:

"We have selected hifiasm and Flye as the default assemblers for various long-read datasets after evaluating several upstream long-read assemblers, including hifiasm [26], Flye [27], Peregrine [40], wtdbg2 [41], IPA [42], HiCanu [43], and Shasta [44]. Our assessment considered their impact on VolcanoSV's downstream large indel SV calling performance. These two assemblers consistently demonstrated superior performance on PacBio Hifi (hifiasm), and for CLR and ONT datasets (Flye), respectively. Additionally, users have the flexibility within this module to choose the most optimal or new assemblers to fulfill their specific requirements."

In the Methods section titled "Haplotype-aware local assembly via augmented phase blocks", we have included the following statement: "Additionally, users have the flexibility within this module to choose the most optimal or new assemblers to fulfill their specific requirements. We also added functionality allowing users to select specific assemblers for target regions defined by a BED file." We have updated our GitHub to reflect these changes.

3. *Figure 3: The caption does not mention that the UpSet plot shows the overlap of true positive SVs detected from the assembly method test, but it should to improve the readability of this section.*

Response: Thanks for pointing out this. We clarified this in the figure legend.

4. *Page 5: The recall ratios for the high-confidence tumor SV calls is quite low. Is there any association of the likelihood of detection with the methods of original discovery used in the original manuscript? For example, is VolcanoSV more likely to detect translocations originally identified by other long read sequence datasets?*

Response: The reviewer’s point is well taken. We analyzed the low recall of complex SVs and added discussion in both the main text and Supplementary Notes.

The revised corresponding main text is provided below:

“Although VolcanoSV outperformed SVIM-asm in detecting complex SVs, its recall was relatively lower compared to its performance with simulated data or indel SVs. To understand the reason behind this low recall and whether VolcanoSV predominantly detects common complex SVs identifiable by other long-read datasets, we analyzed the overlapping VolcanoSV calls between paired normal and tumor HCC1395 samples, and HG002. Our analysis revealed that 1) VolcanoSV failed to detect a number of unique somatic complex SVs in the tumor sample, in addition to the germline SVs; 2) The high-confidence benchmark callset included SV calls only from alignment-based tools, and therefore using it to evaluate VolcanoSV might introduce bias. Overall, it is still challenging to solely use assembled contigs to detect complex SVs due to the limitations of assembly algorithms and the complexity of graph construction. Many genome assembly algorithms build contigs by following the simplest paths through overlapping reads, which may miss complex SVs. These variants create irregular patterns that do not fit into the straightforward paths the algorithms usually prefer or disrupt the continuity of the graph. Detailed results and discussion are outlined in Supplementary Figure 2 and Supplementary Notes 1.1.”

The updated **Supplementary Notes 1.1** and **Supplementary Figure 2** is provided below:

Supplementary Figure 2

Equation 1: $\text{overlap}^N / \text{overlap}^T \approx 1$
 Equation 2: $\text{overlap}^N / \text{HCC1395}^N - \text{overlap}^T / \text{HCC1395}^T > 0$
 Equation 3: $\text{overlap}^N / \text{HG002}^N - \text{overlap}^T / \text{HG002}^N \approx 0$

d

TRA	VolcanoSV	pbsv
Recall	0.32	0.55
Equation 1	0.71	1.01
Equation 2	2.69%	14.41%
Equation 3	-4.87%	0.21%

INV	VolcanoSV	pbsv
Recall	0.29	0.48
Equation 1	1.23	1.31
Equation 2	9.73%	14.14%
Equation 3	5.37%	7.10%

DUP	VolcanoSV	pbsv
Recall	0.13	0.43
Equation 1	1.16	1.14
Equation 2	-0.16%	1.36%
Equation 3	5.75%	2.51%

“Analyzing the low recall of complex SVs in real cancer datasets

To investigate the reason for the low recall or whether VolcanoSV is more likely to detect common complex SVs, which can be identified by other long-read datasets, we analyzed the overlapping VolcanoSV calls between paired normal and tumor HCC1395 samples, and HG002, using PacBio CLR libraries.

In Supplementary Figure 2a (above), $HCC1395^N$ denotes the number of SV calls by VolcanoSV for the HCC1395 normal library, $HG002^N$ for the HG002 library, and $overlap^N$ for the overlapping SV calls between these two call sets. In Supplementary Figure 2b, $HCC1395^T$ denotes the SV calls by VolcanoSV for the HCC1395 tumor library, $HG002^N$ for the HG002 library, $overlap^T$ for the overlapping SV calls between two sets. For an effective tool, we anticipated similar overlap values ($overlap^N \approx overlap^T$); however, there were significantly more calls for HCC1395 tumor library compared to the HCC1395 normal library. To better illustrate this, we formulated three equations (1-3) shown in Supplementary Figure 2c. To achieve high recall with a tool, we expected the following: Equation 1: $overlap^N / overlap^T \approx 1$, indicating the tool called similar number of germline SVs in both normal and tumor samples, part of which could be common SVs shared with HG002; Equation 2: $overlap^N / HCC1395^N - overlap^T / HCC1395^T > 0$, suggesting the tool called a significant number of somatic SVs in addition to the germline SVs in tumor sample; Equation 3: $overlap^N / HG002^N - overlap^T / HG002^N \approx 0$, showing consistent overlap ratios across HG002 samples (similar as Equation 1). Our results showed that the TRA recall (0.32) and INV recall (0.29) by VolcanoSV outperformed the DUP recall (0.13). This aligns with the expected outcomes derived from the three equations for each SV type as shown in Supplementary Figure 2d. Equation 2, which was -0.16% for DUP, suggested that VolcanoSV detected less SVs in tumor samples compared to normal samples (2379 DUPs in tumor versus 2768 DUPs in normal), implying the low recall for DUP.

To create a gold standard complex SV call set, the original paper by Talsania et al. employed various sequencing technologies for both short and long reads, alongside different alignment-based tools to identify SVs. Specifically, for long-read PacBio CLR datasets, they utilized pbsv and Sniffles. We thus expanded our analysis to incorporate pbsv calls, and pbsv did achieve better recall rates for TRA (0.55), INV (0.48), and DUP (0.43) compared to VolcanoSV. This inclusion could help us gain a more comprehensive understanding of the recall dynamics. The metric values derived from the three equations for pbsv supported previous findings, suggesting that these equations can help explain the varying recall levels observed across different tools. Specifically, equation 2 values, which were 14.41%, 14.14%, and 1.36% for TRA, INV, and DUP, respectively, suggested that pbsv detected more unique complex SVs in tumor samples compared to normal samples to achieve

better recall (also indicated by metric values from Equation 1 and 3). However, two important points to consider were: 1) since the high-confidence benchmark callset included SV calls from all alignment-based tools like pbsv, using it to evaluate VolcanoSV might introduce bias; 2) Benchmarking complex SVs is akin to evaluating large indel SVs but may be even more intricate. Evaluation parameters in terms of sequence similarity and breakpoints would greatly impact the recall of a specific tool.

In conclusion, the metric values derived from the three equations partially offer an explanation for low recall rates in complex SVs by VolcanoSV. To improve recall, it is crucial to identify more unique calls in tumor samples that distinctly differ from those in normal samples. However, it is challenging to solely use assembled contigs to detect complex SVs. This could be due to the limitations of assembly algorithms and the complexity of graph construction. Many genome assembly algorithms are designed to create contigs by tracing the simplest paths through a graph of overlapping reads, such as an overlap graph, constructed from long reads. However, these algorithms typically prioritize the most common paths, which can overlook complex structural variants like translocations. Translocations rearrange DNA by moving segments from one part of the genome to another, creating non-linear or non-sequential patterns. These patterns disrupt the graph's continuity and form connections that are challenging to interpret correctly because they deviate from the usual linear arrangement of chromosomes, complicating the detection of such variants.”

5. *Page 5: I feel that the comparison of assembly- and alignment-based SV callers is incomplete. The authors appear to try to highlight the strengths of assembly-based methods by only using their alignment-based callsets in a parameter-optimization exercise to demonstrate the inability of these tools to generate precise breakpoints. However informative this may be, it is not a fair demonstration. I would prefer to see F1 and Recall statistics from the alignment-based methods on the same GIAB, simulation, and tumor datasets to appropriately compare the results of this test.*

Response: The reviewer’s point is well taken. The initial aim of this analysis was to emphasize that assembly-based SV calls offer superior resolution in terms of sequence and breakpoints compared to alignment-based calls, achieved by changing different ranges of SV evaluation parameters during benchmarking. However, we acknowledge that merely highlighting this comparison is incomplete, as there are trade-offs between assembly-based and alignment-based calls. In alignment with the paper's focus on diploid assembly and assembly-based calls, as

well as the reviewer's previous general comments, we dropped this comparison and revised the corresponding text in the Results section.

6. *Page 7: The T2T assemblies often have assembled heterochromatin regions that are difficult to assemble apart from the use of specialized graph-based methods. Are the false positive calls from the assembly-based tools localized to some of these heterochromatin regions? Is that where the main source of error exists?*

Response: The reviewer raises a valid point. We performed additional analysis for the false positive calls. The main source of error did locate in the heterochromatin regions like centromeres and telomeres.

The **corresponding Results section** and **Supplementary Figure 6** are provided below:

“We conducted additional analysis on the falsely discovered SVs based on the T2T-CHM13 reference. The results, illustrated in the UCSC Genome Browser (Supplementary Figure 6-7) and Table 3, showed that 92.6%, 96.4%, 94.2%, and 92.7% of FD for Hifi_L6 by VolcanoSV, PAV, SVIM-asm, and Dipcall, respectively, were located in heterochromatin regions like centromeres and telomeres. For Hifi_L7, the percentages were lower, with 79.9%, 85.8%, 81.8%, and 82.4% of SVs identified by VolcanoSV, PAV, SVIM-asm, and Dipcall, respectively, also located in these complex regions. Notably, the majority of FDs were concentrated in chromosomes 1, 7, 9, 15, and 16. Heterochromatin regions pose challenges for assembly, often requiring specialized graph-based methods for accurate resolution.”

Supplementary Figure 6: False discovery analysis in Hifi L6 on the T2T-CHM13 reference. The UCSC Genome Browser displays six tracks for chromosomes 1 to 22: “CenSat Annotation” for centromere annotation, “Telomere” for telomere annotation, and individual tracks for false discoveries identified on T2T-CHM13 by VolcanoSV, PAV, SVIM-asm, and Dipcall in Hifi L6, labeled as “VolcanoSV FD Hifi L6”, “PAV FD Hifi L6”, “SVIM-asm FD Hifi L6”, and “Dipcall FD Hifi L6”, respectively.

Supplementary Figure 6 (next page):

UCSC Genome Browser on Human Jan. 2022 (T2T CHM13v2.0/hs1) (hs1)

We have also included a Methods section titled “Annotating and visualizing false discoveries on the T2T-CHM13 reference” for this analysis.

7. Page 7: One of the distinct advantages of alignment-based variant calling methods is that they often have lower computational footprints. A discursive analysis is not required here, but what is the relative difference in computational time and resources of Volcano-SV vs alignment based methods of SV discovery?

Response: The reviewer raises a valid point. Assembly-based methods usually require more computational resources due to the complexities involved in the assembly process, in contrast to the alignment process required by alignment-based methods. We have expanded our analysis and discussion of this aspect in the "Computational Costs" Results section.

8. *Github repository: The authors include an unlinked version of HifiASM in their repository. Proper credit to the original authors of the repo should be given, or a hard link to that specific version of the repo should be added to the main repository for VolcanoSV.*

Response: Thank you for bringing this to our attention. We have now included the link for hifiasm and other assemblers in our GitHub repository, as well as the credit within the corresponding section of our code.

Reviewer #1 (Remarks on code availability):

The code base has a suitable README and is documented with some quick start instructions. It seems suitable to test and run the software package as presented. I note some discrepancies in code reuse from other software packages in my comments to the Authors.

Response: Thank you for reviewing our code and readme. We have fixed the issue and added more functionality and documentation based on your feedback.

Reviewer #2 (Remarks to the Author):

The manuscript introduces VolcanoSV, a novel hybrid SV detection pipeline that enhances the precision of identifying structural variants (SVs) in the human genome. The tool combines reference genome and local de novo assembly methods to accurately pinpoint SV breakpoints and sequences. Despite its promising potential, the manuscript requires significant revision to address certain issues and improve clarity and completeness in presenting VolcanoSV's methodology and findings.

We thank Reviewer 2 for the positive evaluation towards the potential of this new method and for the insightful comments on this study. We have performed additional analyses to further improve the rigor of this method. All the changes are highlighted in red in the main manuscript and supplementary information.

Below are the major points that need to be addressed:

- 1. VolcanoSV incorporates existing tools for sequence assembly, which may not always be the most suitable choice. Particularly in regions rich in segmental duplications, it is recommended to employ tools that outperform Flye in handling such complexities.*

Response: The reviewer's point is well taken. We have included a Results section titled "Selecting assemblers for regions enriched in segmental duplications" and a Methods section titled "Assessing diploid assembly using Flagger and annotating regions enriched in segmental duplications" to address this.

In the Methods section titled "Haplotype-aware local assembly via augmented phase blocks", we have included a statement as below:

"Additionally, users have the flexibility within this module to choose the most optimal or new assemblers to fulfill their specific requirements. We also added functionality allowing users to select specific assemblers for target regions defined by a BED file."

The **corresponding Results section** is provided below:

"Selecting assemblers for regions enriched in segmental duplications"

We have selected hifiasm and Flye as the default assemblers for various long-read datasets after evaluating several upstream long-read assemblers, including hifiasm [26], Flye [27], Peregrine [40], wtdbg2 [41], IPA [42], HiCanu [43], and Shasta [44]. Our assessment considered their impact on VolcanoSV's downstream large indel SV calling performance. These two assemblers consistently demonstrated superior performance on PacBio Hifi (hifiasm), and for CLR and ONT datasets (Flye), respectively. However, users also have the flexibility within the assembly module in VolcanoSV to choose the most appropriate or new assemblers to fulfill their specific requirements. For example, when assembling regions enriched in segmental duplications (SDs), these two assemblers may not be the most suitable choice. To investigate the performance of different assemblers in SD-enriched regions, we employed Flagger [45] to detect misassemblies enriched in SDs in three representative libraries. Flagger is a read-based pipeline that maps long reads to the

phased diploid assembly in a haplotype-aware manner. It identifies coverage inconsistencies within these read mappings that are likely due to assembly errors. Details are provided in the Methods section.

For ONT_L1, we additionally employed wtdbg2, Shasta, and NextDenovo [46] in VolcanoSV to perform local assembly for all phase blocks. Using Flagger, we first identified collapsed components in diploid assembly by VolcanoSV using different assemblers. These collapsed components represent regions with additional copies in the underlying genome that have been collapsed into a single copy, indicating potential misassemblies for SDs. We then further assessed the SD reliability of the Flagger annotation for collapsed components. To achieve this, we aligned the collapsed components from all diploid assemblies to the GRCh38 reference genome and intersected them with the SD annotations for HG002 based on GRCh38. We then calculated and compared the total length of SD annotation regions that overlap with collapsed regions by Flagger across all assemblies. The results demonstrated that VolcanoSV assembly using Flye, wtdbg2, Shasta, and NextDenovo generated 34.9MB, 47.6MB, 34.2MB, and 22.7MB of collapsed components, respectively, which suggested collapsed SDs.

For CLR_L1, we additionally employed wtdbg2, and NextDenovo in VolcanoSV to perform local assembly for all phase blocks. The results demonstrated that VolcanoSV assembly using Flye, wtdbg2, and NextDenovo generated 28.8MB, 32MB, and 30.2MB of collapsed components, respectively, which suggested collapsed SDs. For Hifi_L1, we additionally employed Hicanu in VolcanoSV to perform local assembly for all phase blocks. The results demonstrated that VolcanoSV assembly using hifiasm and Hicanu generated 17.4MB and 15.7MB of collapsed components, respectively, which suggested collapsed SDs.

These results suggested that NextDenovo in VolcanoSV assembled fewer collapsed regions than Flye in ONT_L1, indicating its potential superiority for regions enriched in SDs. Similarly, Hicanu in VolcanoSV outperformed hifiasm in Hifi_L1. Notably, Flye in VolcanoSV assembled the fewest collapsed regions in CLR_L1 compared to other assemblers.”

2. The validation scope of the method is somewhat limited, and the metrics obtained for some competitors are very similar, with differences only evident after

the second decimal place. The manuscript would benefit from demonstrating the implications of such improvements, for example, showing how a 0.59% increase in GT accuracy translates into tangible benefits or impacts on downstream analyses. The Truvari toolkit should be more thoroughly described, with its parameters clearly explained to enhance understanding.

Response: The reviewer's point is well taken. We have added a Results section titled "SV annotation", Table S4 and Supplementary Figure 1 to show the implication of such improvement in all 14 libraries.

The **corresponding Results section** is provided below:

"SV annotation

Although VolcanoSV further improved F1 scores and genotyping accuracy compared to the second-ranked tool in each dataset, potential benefits or impact of this improvement on downstream analyses remained unclear. To delve deeper into these unique true positive (TP) SVs with correct genotypes (GT) identified by VolcanoSV compared to the second-ranked tool, we annotated the SVs with their predicted effects on other genomic features using the Ensembl Variant Effect Predictor (VEP) [31].

For example, we extracted and annotated 300 unique TP SVs with correct genotypes identified by VolcanoSV compared to the second-ranked tool, SVIM-asm, in Hifi_L1. These SVs overlapped with 258 genes, 939 transcripts, and 120 regulatory features, including 266 novel SVs and 34 existing ones. Among these, 125 SVs were coding sequence variants and 16 were in-frame deletions. More information on the consequences of these SVs is demonstrated in Supplementary Figure 1. Additionally, we performed an exact match comparison of these SVs in terms of sequence and breakpoints with those in the Genome Aggregation Database (gnomAD) [32]. We identified 29 matching SVs, 12 of which are rare variants with an allele frequency (AF) of less than 1%. Furthermore, we found that these SVs allowed an additional 85 genes to be phased by VolcanoSV. Genes are considered phased if all heterozygous variants within it are phased.

We performed these analyses for all 14 long-read datasets (Table S4 and Supplementary Figure 1), and the results suggested that the improvement by VolcanoSV over the second-ranked tool in each dataset provided a substantial number of SVs with potential phenotypic effects."

For the Truvari toolkit and its parameters, we have included more description in the first Results section titled “VolcanoSV exhibits exceptional performance across 14 different real long-read datasets” and the corresponding Methods section titled “Benchmarking SV calls using Truvari”.

3. *The methodology is not clearly described. The notation is rather unclear, and inconsistent font usage further complicates readability. What is the null hypothesis in the chi-square test used in unphased reads assignment? Is the similarity defined in equation (4) a metric? On what basis were the thresholds selected in equation (7) (13) (14) (19) and (20) ? The sentence: “In an ideal case, when the subcluster only contains one signature, this signature is directly selected as the actual SV call” should be accompanied by the statistics of how this looks in general. In equation (12), it is unclear why the absolute value pertains to only one sequence, and additionally, the parentheses are mismatched.*

Response: The reviewer’s point is well taken.

1) We added more clarification for notations and used consistent font based on the NC journal requirements.

2) The null hypothesis (H0) is as follows: the normalized similarity metric between the unphased read and the candidate PS_HP is not significantly different from what would be expected by random chance, i.e., $Norm.Sim_{(r)} \leq Q_{s\%}(X)$. However, since our process involves comparing a normalized similarity metric to a quantile-based threshold rather than using observed vs. expected frequencies (as in a traditional chi-square test), we renamed our significance test to be “Empirical Distribution Quantile-Based Test”. We revised the corresponding Methods section.

3) Yes, the normalized similarity is a metric. For each unphased read, we compare its normalized similarity metric to the cut-off threshold $Q_{s\%}(X)$. If the metric is higher than the cut-off, it is considered significant, suggesting a potential association with the corresponding PS_HP. If a normalized similarity metric for an unphased read does not exceed the cut-off, we fail to reject the null hypothesis for that specific read and candidate PS_HP combination. This implies that there is no significant association, and the observed similarity might be due to random chance.

The revised corresponding **Methods section** for **2)** and **3)** is provided below:

“VolcanoSV utilizes an empirical distribution quantile-based significance test to evaluate the normalized similarity metrics between unphased reads and candidate PS_HPs. A level (r) (10% by default) is used, and the cut-off threshold for

significance is the $(1-r)$ quantile of the normalized similarity vector χ . Metrics exceeding this threshold are considered significant, and reads are assigned accordingly. The null hypothesis (H_0) posits that the normalized similarity metric between the unphased read and the candidate PS_HP is not significantly different from what would be expected by random chance, i.e., $\text{NormSim}\{\} \leq Q_{1-r}(\chi)$. For each unphased read, we compare its normalized similarity metric to the cut-off $Q_{1-r}(\chi)$. If the metric is higher than the cut-off, it is considered significant, suggesting a potential association with the corresponding PS_HP. Conversely, if a normalized similarity metric for an unphased read does not exceed the cut-off, we fail to reject the null hypothesis for that specific read and candidate PS_HP combination, implying that there is no significant association, and the observed similarity might be due to random chance.”

4) In the revised version, we have introduced several threshold parameters for these metric equations and provided justifications for the selection of these thresholds.

5) Thanks for pointing out this. We clarified and revised this part. In general, the majority of subclusters constituted the ideal case. In this step, we tried to remove signature redundancy for each SV on each haplotype. In most cases, the subcluster contains only one signature, which is directly selected as the actual SV call. In the few cases where a subcluster contains more than one signature, VolcanoSV selects the signature with the largest length as the actual SV call.

6) Thanks for pointing out this. The absolute sign was removed since “svlen2” is a positive value. We fixed the formula.

4. The figures are very unclear. Fig. 1 should be divided into several parts, while Figure 2 does not add any value and could be presented as a table instead.

Response: The reviewer’s point is well taken. We have split Figure 1 into two figures (Figure 1-2) and refined the pipeline for better visualization. Figure 2 has been updated to Figure 3 in the revised version. Although we have kept Figure 3 we also present the values in detail in Table 2.

Reviewer #2 (Remarks on code availability):

This GitHub resource offers clear documentation, commented code, and practical examples, making it accessible for developers.

Response: Thank you for reviewing our code and readme. We have added more functionality and documentation based on your feedback.

REVIEWERS' COMMENTS

Reviewer #1 (Remarks to the Author):

Summary: The authors' responses are both thoughtful and thorough. All of my major points were addressed in this revision, and I only have one remaining minor point that will need to be addressed:

Discussions: in light of the additional testing, I would like to see one or two sentences that provide a recommendation or use-case for Volcano-SV. Alternatively, would the authors propose using Volcano-SV as part of a comprehensive workflow or a refinement tool?

Reviewer #1 (Remarks on code availability):

The code is now suitable and makes proper references to external sources.

Reviewer #2 (Remarks to the Author):

Following a second review of the manuscript titled "VolcanoSV enables accurate and robust structural variant calling in diploid genomes from single-molecule long read sequencing," I find that the authors have addressed most of the previous comments and concerns. They have also added further validation experiments, which have strengthened the results. However, in my opinion, the notation used in the paragraphs "Contig alignment-based signature collection for large indel SV detection" and "Clustering of large indel SV signatures on each haplotype" remains somewhat unclear. Overall, the manuscript is now acceptable and can be published.

Response to Reviewers

We appreciate the reviewers' careful reading of our revised manuscript and have addressed the remaining minor comments.

Reviewer #1:

Summary: The authors' responses are both thoughtful and thorough. All of my major points were addressed in this revision, and I only have one remaining minor point that will need to be addressed:

Discussions: in light of the additional testing, I would like to see one or two sentences that provide a recommendation or use-case for Volcano-SV. Alternatively, would the authors propose using Volcano-SV as part of a comprehensive workflow or a refinement tool?

Response: Thanks for your positive feedback overall! This point is well taken. We recommend using VolcanoSV as a comprehensive workflow. In the Discussion section, we have added the following recommendation statement:

"VolcanoSV is a comprehensive workflow, and we recommend users to use it directly to generate diploid assembly and detect variants. However, the ``VolcanoSV-asm" component can be used independently to generate diploid assembly inputs for other assembly-based tools. Additionally, the ``VolcanoSV-vc" component can be independently used to detect variants by taking any diploid assembly as input from other assembly tools."

Remarks on code availability:

The code is now suitable and makes proper references to external sources.

Reviewer #2:

Following a second review of the manuscript titled "VolcanoSV enables accurate and robust structural variant calling in diploid genomes from single-molecule long read sequencing," I find that the authors have addressed most of the previous comments and concerns. They have also added further validation experiments, which have strengthened the results. However, in my opinion, the notation used in the paragraphs "Contig alignment-based signature collection for large indel SV detection" and "Clustering of large indel SV signatures on each haplotype" remains somewhat unclear. Overall, the manuscript is now acceptable and can be published.

Response: Thanks for your positive feedback overall. To address the point, we have made some additional edits to this section for clarity and refer to Figure 2, which provides a graphical illustration of the methods, likely making them easier to understand.